# Behavioral representational similarity analysis reveals how episodic learning is influenced by and reshapes semantic memory

Catherine R. Walsh [1] ✉ & Jesse Rissman [1,2,3,4]

While semantic and episodic memory have been shown to influence each other, uncertainty remains as to how this interplay occurs. We introduce a behavioral representational similarity analysis approach to assess whether semantic space can be subtly re-sculpted by episodic learning. Eighty participants learned word pairs that varied in semantic relatedness, and learning was bolstered via either testing or restudying. Next-day recall is superior for semantically related pairs, but there is a larger benefit of testing for unrelated pairs. Analyses of representational change reveal that successful recall is accompanied by a pulling together of paired associates, with cue words in semantically related (but not unrelated) pairs changing more across learning than target words. Our findings show that episodic learning is associated with systematic and asymmetrical distortions of semantic space which improve later recall by making cues more predictive of targets, reducing interference from potential lures, and establishing novel connections within pairs.

Despite early theories that proposed a psychological and neurobiological separation between semantic and episodic memory systems[1,2], there is an increasing body of work that suggests the two systems are more intertwined than previously believed[3,4]. Neuroimaging experiments have demonstrated shared neural activation[5] and functional connectivity[6,7] during episodic and semantic memory processes, and pre-existing semantic knowledge can act as a scaffold to facilitate the acquisition of new episodic memories[8–10]. Moreover, semantic relatedness has been shown either facilitate[11–16] or impair[17,18] episodic memory performance, depending on factors such as recall delay, degree of relatedness within the to-be-learned pairs, and the semantic relatedness of the broader stimulus set[15]. Episodic experiences can also influence semantic knowledge by integrating new information as learning occurs, or by emphasizing task or context-relevant semantic features in pre-existing semantic space[19–21]. However,

further specification of the mechanisms of these putative bidirectional episodic/semantic interactions is needed.

One common assessment of episodic memory involves presenting pairs of items and later probing retention of the associations. Although one-shot learning of paired associates is possible, many paradigms have participants with re-engage with the material through retrieval practice or restudying, and there is a well-established benefit of the former, known as the testing effect[22–28]. There is debate as to whether the "desirable difficulty"[29,30] or effortfulness[31] of searching for and retrieving a target association is what strengthens memory or whether testing is advantageous because the episodic experience of retrieval practice is more contextually similar to the final test[32].

While researchers have increasingly acknowledged the interdependence of episodic and semantic memory, there are relatively few studies of the testing effect that directly manipulate the semantic information within to-be-learned pairs of items[28] or integrate its role

[1]Department of Psychology, University of California, Los Angeles, CA, USA. [2]Department of Psychiatry & Biobehavioral Sciences, University of California, Los Angeles, CA, USA. [3]Brain Research Institute, University of California, Los Angeles, CA, USA. [4]Integrative Center for Learning and Memory, University of California, Los Angeles, CA, USA. ✉e-mail: crewalsh@g.ucla.edu

into mechanistic accounts. Carpenter[33] proposed that retrieving information from memory necessitates elaborative processes that induce spreading activation to semantically related information[34,35], which can provide additional retrieval cues[31]. Consistent with this framework, one recent study showed that when to-be-learned images do not contain meaningful semantic information, there is no benefit for retrieval practice compared to restudying the images[36]. A separate account suggests that testing supports memory by facilitating semanticisation[37] (i.e. a shift towards more generic semantic representations as opposed to detail-rich episodic representations) and relational processing, which promotes attention to semantic information[38].

The degree to which to-be-learned items have a pre-existing semantic relationship may influence how they are associated in memory. The episodic binding of two items need not be symmetrical, in the sense that the ability of item A to predict item B does not necessarily equate with the ability of item B to predict item A. For instance, when pairs of words are learned in one direction (cue word A→target word B), the act of testing an unrelated pair in the forward direction (A→?) also improves associative memory in the reverse direction (B→?), yet when related pairs are tested in the forward direction (A→?), it does not improve recall of the reverse direction (B→?)[39,40]. Recent neuroimaging work has also shown asymmetrical integration of associative pairs[41]. For example, when novel faces are paired with famous faces, the neural representation of the novel face becomes more similar to the representation of the paired famous face, which itself shows minimal representational change. In contrast, when a novel face is paired with another novel face, the neural representations of the two faces become more similar, but change equally.

One neurobiologically-inspired computational modeling account of associative learning known as the non-monotonic plasticity hypothesis (NMPH) attempts to explain the testing effect and account for the role of semantic information through the relative co-activation of to-be-learned items and the associated representational change. This framework proposes that changes in memory strength are driven by the relative activation of items, such that memory for items that are strongly co-activated is strengthened, while items that are moderately co-activated are weakened or differentiated[42,43]. When paired items are restudied and brought to mind together, they are strongly co-activated, and thus strengthened[44]. When paired items undergo retrieval practice, there is also strong co-activation, but because retrieval is often imprecise, it will also tend to moderately co-activate semantically related concepts[34,35]. According to the NMPH, this moderate activation suppresses memory for the related items and differentiates the target to reduce interference and strengthen memory more than restudying[42,43,45–48].

In the present preregistered study, we sought to investigate the influence of semantic relatedness on the testing effect and understand how episodic paired associate learning might sculpt pre-existing semantic space. We had participants learn semantically related and unrelated pairs of words via testing or restudying and assessed their memory the next day. Although we were interested in how cued recall accuracy would vary depending on semantic relatedness and learning condition, our primary focus was on whether and how the semantic representations of the words changed over learning. For this, we developed a behavioral representational similarity analysis approach, which we applied to data from a similarity-based word arrangement task that participants performed before and after learning. This allowed us to investigate the bidirectional interaction of episodic learning and semantic knowledge by indexing changes in the associative structure and semantic representation of individual words. Given the existing computational modeling work and literature on the role of semantic information, we expected to see an overall memory benefit for semantically related pairs. We thus predicted that these already-advantaged pairs would have less to gain from testing than

unrelated pairs. We anticipated that tested pairs would undergo more representational change, and that the amount of representational change would be correlated with behavioral performance. Finally, we expected to see asymmetric change in the semantic structure of related pairs, where representations of targets would get drawn towards those of the cues, and symmetric changes for unrelated pairs of words.

In this work, we show that the testing effect is reduced for semantically related pairs of words due to the relative improvement in recall of restudied pairs. We also show that when pairs lack a prior semantic relationship, testing is necessary to induce representational change and that this change draws cues and targets together symmetrically. Testing also weakens the relationship of the cue words with other moderately related non-target associates. In contrast, prior semantic knowledge can rescue restudied pairs by inducing asymmetric representational semantic change, which makes cues more predictive of their associated targets. Finally, we show that the relationship between representational change and recall accuracy of word pairs depends on the interaction of word position, semantic relatedness, and learning condition, where greater representational change in the cue is associated with better recall, regardless of learning condition or semantic relatedness, but representational change in the target is only associated with better recall when pairs are semantically unrelated and tested.

## Results
### Recall accuracy
We began by probing whether recall accuracy for the targets of each cue-target pair systematically varied based on the semantic relatedness (related vs. unrelated) and the learning condition (testing vs restudying, following two initial exposures); Fig. 1.

On Day 1, we could only assess recall accuracy for tested pairs, since performance for restudied pairs merely reflected participants' ability to type the visible target word. As expected, semantically related word pairs were recalled better than semantically unrelated pairs ($t_{(79)} = 9.979$, $p < 0.001$, $d = 1.12$, 95% CI = [0.19 0.28]); Fig. 2A. After a 24-hour delay (Day 2), a RM-ANOVA revealed significant main effects of semantic relatedness ($F_{(1,79)} = 362.227$, $p < 0.001$, $\eta_G^2 = 0.462$) and learning condition ($F_{(1,79)} = 25.240$, $p < 0.001$, $\eta_G^2 = 0.045$) but no statistically significant interaction ($F_{(1,79)} = 0.076$, $p = 0.78$, $\eta_G^2 = 9.94 \times 10^{-5}$); Fig. 2B. Comparison of marginal means at the final test that related pairs (M = 0.629, SD = 0.183) had a higher probability of recall than unrelated pairs (M = 0.281, SD = 0.202), and tested pairs (M = 0.496, SD = 0.258) were more likely to be recalled than restudied pairs (M = 0.415, SD = 0.256).

Given that participants were provided with no feedback, it is possible that tested pairs that were not successfully retrieved on Day 1 would not benefit from testing, potentially obscuring an interaction between semantic relatedness and learning condition on Day 2. To investigate this, tested pairs were split into those that were correctly recalled at initial learning and those that were not, revealing a significant relatedness by learning condition interaction for Day 2 recall performance ($F_{(2,154)} = 23.531$, $p < 0.001$, $\eta_G^2 = 0.054$); Fig. 2C, D. Follow up paired t-tests revealed significant testing effects (i.e. the contrast of pairs that were tested and correctly recalled at Day 1 versus pairs that were restudied) for both related pairs ($t_{(79)} = 9.575$, $p < 0.001$, $d = 1.070$, 95% CI = [0.187 0.285]) and unrelated pairs ($t_{(79)} = 12.361$, $p < 0.001$, $d = 1.382$, 95% CI = [0.283 0.391]), with a larger effect of learning condition for unrelated pairs. Pairs that were tested but recalled incorrectly at Day 1 showed significantly lower accuracy on Day 2 than both restudied pairs (related: $t_{(77)} = 13.602$, $p < 0.001$, $d = 1.540$, 95% CI = [0.354 0.476]; unrelated: $t_{(79)} = 9.692$, $p < 0.001$, $d = 1.084$, 95% CI = [0.149 0.226]) and tested pairs that were recalled correctly at Day 1 (related: $t_{(77)} = 22.293$, $p < 0.001$, $d = 2.524$, 95% CI = [0.543 0.710]; unrelated: $t_{(79)} = 18.870$, $p < 0.001$, $d = 2.110$, 95% CI = [0.469 0.580]). These results indicate that semantic relatedness reduces the

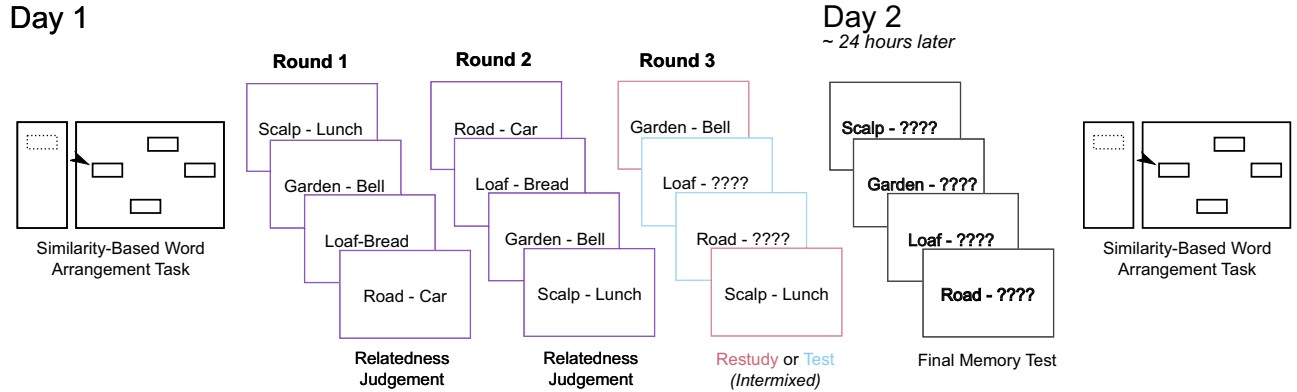

**Fig. 1 | Procedural overview.** The experiment took place over two days. Prior to learning, participants performed the Similarity-based Word Arrangement Task (SWAT), where they rated the similarity of subsets of words across four trials (60 words per trial). Next, participants had three opportunities to learn 60 pairs of words. During the first two opportunities (Rounds 1 and 2), participants made judgements about the relatedness of the words within the pair. For the third opportunity (Round 3), pairs were either restudied (illustrated here with maroon border) or tested (blue border). On Day 2, participants completed a final cued recall test for all learned pairs, followed by another four trials of the SWAT.

magnitude of the testing effect by improving recall of related restudied pairs. The benefit of semantic relatedness overcomes the relatively ineffective learning method of restudying, leaving less to gain by testing.

### Change in pairwise representational similarity

While differences in recall accuracy based on semantic relatedness and learning condition show that these factors are consequential for memory, they cannot show how this happens. To gain mechanistic insight, we turned to the changes in pairwise representational similarity (measured by the difference between within pair similarity at the final and initial Similarity-Based Word Arrangement Task (SWAT) assessments; Fig. 3), which provides a more direct measurement of how the semantic representations of our word set change across learning; Fig. 4.

First, we ran a linear mixed-effects model (LMM) testing whether there were differences in the change in similarity for pairs that were correctly recalled at Day 2 relative to those incorrectly recalled at Day 2 and to a control condition of random word pairings that were never experienced during learning. Pairs that were correctly recalled at Day 2 changed more than those that were incorrectly recalled ($t_{(158)} = 2.566$, $p = 0.023$, $d = 0.20$, 95% CI = [$1.84 \times 10^{-5}$ $6.29 \times 10^{-4}$]) and random pairs ($t_{(158)} = 3.112$, $p = 0.007$, $d = 0.25$, 95% CI = [$8.73 \times 10^{-5}$ $6.98 \times 10^{-4}$]), but the change in pairs that were incorrectly recalled at Day 2 was not statistically significantly different from that of random pairs ($t_{(158)} = 0.547$, $p = 0.585$, $d = 0.04$, 95% CI = [$-3.74 \times 10^{-4}$ $2.36 \times 10^{-4}$]); Fig. 5A.

We next ran a series of one-sample t-tests (with Holm-Bonferroni corrections for multiple comparisons) to determine whether change in similarity in our conditions of interest was significantly different from zero; Fig. 5B. For this analysis (and all hereafter), we opted to exclude tested pairs that were incorrectly recalled at Day 1 because they did not incur the benefit of testing. Significant changes in similarity were observed for related pairs that were correctly recalled at Day 2, regardless of learning condition (tested: $t_{(79)} = 3.788$, $p = 0.002$, $d = 0.423$, 95% CI = [$3.045 \times 10^{-4}$ $9.804 \times 10^{-4}$]; restudied: $t_{(79)} = 4.258$, $p < 0.001$, $d = 0.476$, 95% CI = [$3.763 \times 10^{-4}$ $1.037 \times 10^{-3}$]), and unrelated pairs that were tested and correctly recalled at Day 2 ($t_{(74)} = 3.085$, $p = 0.017$, $d = 0.356$, 95% CI = [$2.736 \times 10^{-4}$ $1.272 \times 10^{-3}$]). All other comparisons were not statistically significantly different from zero ($p$ values > 0.1; Supplementary Table 2). When change in similarity across conditions was analyzed in an LMM with fixed effects of relatedness, learning condition, and final recall success, there were no main effects or interactions between relatedness and learning condition;

there was, however, a significant main effect of final recall success ($t_{(528)} = 1.965$, $p = 0.050$, $\eta_p^2 = 0.0073$, 95% CI = [$1.348 \times 10^{-6}$ $6.372 \times 10^{-4}$]), where pairs that were correctly recalled at Day 2 showed significantly more change in similarity than those that were not.

Although correctly recalled pairs showed the most overall representational change, it is also possible that there might be changes within the local semantic neighborhoods of the learned cue words that reflect the repulsion of potential competitor words (i.e. potential lures) to reduce interference; Fig. 4A. To test for this, we first characterized the strength of potential lures for each cue word using the LSA cosine similarity between the cue word and all other words in our 120-word set. For example, for the pair GENDER – FEMALE, the word MOTHER might interfere with recall, while CAVERN likely would not. We then calculated the change in similarity across learning for cues in successfully recalled to-be-learned pairs and their potential lures (e.g., GENDER – MOTHER). We used this change in similarity as the outcome variable of an LMM with fixed effects of relatedness, learning condition, and lure strength and random effects of relatedness and learning condition. This model showed a significant learning condition by lure strength interaction ($t_{(248400)} = 2.840$, $p = 0.005$, $\eta_p^2 = 3.25 \times 10^{-6}$, 95% CI = [$-3.130 \times 10^{-4}$ $-5.740 \times 10^{-5}$]); Fig. 5C. Follow up t-tests revealed that very strong lures are drawn together more than weak/non-lures when they were associated with both tested ($z = 6.029$, $p < 0.001$, $d = 0.12$, 95% CI = [$4.90 \times 10^{-4}$ $1.22 \times 10^{-3}$]) and restudied pairs ($z = 6.689$, $p < 0.001$, $d = 0.15$, 95% CI = [$6.59 \times 10^{-4}$ $1.48 \times 10^{-3}$]). In contrast, moderate lures associated with tested pairs were pulled together less than tested weak/non-lures ($z = 4.182$, $p < 0.001$, $d = 0.03$, 95% CI = [$7.48 \times 10^{-5}$ $3.13 \times 10^{-4}$]). Because there was generalized semantic change even for weak/nonlures (see Supplementary Note 3), we further probed this interaction by contrasting each lure bin with the weak/nonlures across learning condition. This analysis revealed that moderate lures associated with tested pairs are drawn together less than those associated with restudied pairs ($z = 2.840$, $p = 0.014$, $d = 0.03$, 95% CI = [$5.740 \times 10^{-5}$ $3.129 \times 10^{-4}$]). All other baseline-corrected comparisons were not statistically significant ($p$ values > 0.05); these comparisons, in addition to pairwise comparisons between other lure bins and other significant effects, are reported in Supplementary Note 3.

These results show that successful recall not only pulls to-be-learned word pairs closer together in representational space, but also sculpts the overall representational space by drawing highly similar words closer to the cue word to potentially serve as additional retrieval cues for the to-be-learned target. Testing additionally repels moderate lures that are unlikely to serve as retrieval cues and could potentially interfere with successful recall.

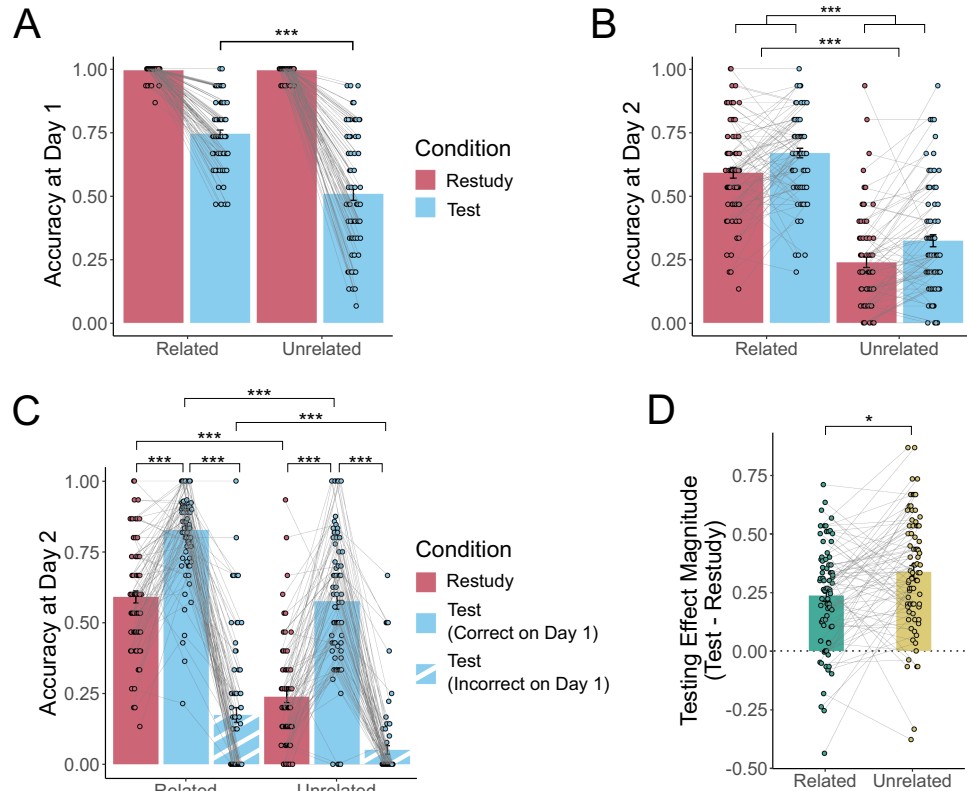

**Fig. 2 | Retrieval accuracy across days as a function of semantic relatedness and learning manipulation. A** On Day 1, a two-tailed t-test revealed that pairs that underwent testing during Round 3 showed a significant effect of relatedness (blue bars; $t_{(79)} = 9.979$, $p < 0.001$, $d = 1.12$, 95% CI = [0.19 0.28]). Restudied pairs from Day 1 are shown (maroon bars) but were not analyzed since the Day 1 restudy score merely reflects the ability to re-type the target word that was displayed on the screen, rather than memory strength. **B** On Day 2 (our critical measure of learning outcomes), a RM-ANOVA revealed that there was better cued recall of target words from related pairs than unrelated pairs ($F_{(1,79)} = 362.23$, $p < 0.001$, $\eta_G^2 = 0.462$), and better recall for tested pairs (blue bars) than restudied pairs (maroon bars) ($F_{(1,79)} = 25.24$, $p < 0.001$, $\eta_G^2 = 0.450$), but no statistically significant interaction. **C** Splitting tested word pairs based on whether or not they were successfully recalled on Day 1 reveals a significant semantic relatedness by learning

manipulation interaction ($F_{(2,154)} = 23.53$, $p < 0.001$, $\eta_G^2 = 0.045$), showing a larger testing effect (i.e. an advantage for tested pairs that were correctly recalled at Day 1 over restudied pairs) for unrelated pairs than for related pairs. Solid blue bars reflect tested pairs that were recalled correctly at Day 1, striped bars reflect tested pairs that were not recalled correctly at Day 1. **D** The interaction was driven by a larger testing effect (i.e. an advantage for tested pairs that were correctly recalled at Day 1 over restudied pairs) for unrelated pairs (yellow) than for related pairs (green). Across all panels, open circles reflect means of individual participants ($N = 80$), with connecting lines showing within-subject differences across conditions. Error bars reflect standard error of the mean. Symbols reflect statistically significant differences across conditions using Holm-Bonferroni corrections for multiple comparisons (*$p < 0.05$, **$p < 0.01$, ***$p < 0.001$).

## Change in overall representational similarity structure

A complementary approach to our analyses of the relationship of words within a to-be-learned pair is to investigate how the semantic relationship of each word changes with respect to all other words in the set. To explore this, we extracted from the full representational similarity matrix the row vector reflecting a word's similarity to its 20 nearest semantic neighbors and compared this across learning; Fig. 4B. For example, the representation of GENDER can be defined by its similarity to its nearest semantic neighbors, including CHILDREN, MOTHER, TEACHER, and PARENT. By comparing the similarity of GENDER to each of these words across learning, we can quantify how much the representation of GENDER changes. When Fisher z-transformed correlation values were entered into an LMM with fixed effects of relatedness, learning condition, final recall success, and word position (cue vs target), and random effects of word position and learning condition, there was a significant relatedness by word position by final recall success interaction ($t_{(967)} = 2.607$, $p = 0.009$, $\eta_p^2 = 0.007$, 95% CI = [−0.206 −0.030]), Fig. 6A, in addition to significant relatedness by final recall success interaction ($t_{(967)} = 1.986$, $p = 0.047$, $\eta_p^2 = 0.0041$, 95% CI = [0.0014 0.153]), and position by final recall success interaction ($t_{(964)} = 2.581$, $p = 0.009$, $\eta_p^2 = 0.0068$, 95%

CI = [0.024 0.175]). Additionally, there was a main effect of final recall success ($t_{(963)} = 2.660$, $p = 0.007$, $\eta_p^2 = 0.0073$, 95% CI = [−0.135 −0.021]). Follow-up t-tests revealed that for related pairs that were successfully recalled at Day 2, target words underwent more learning-induced representational change than cue words ($t_{(372)} = 3.546$, $p = 0.002$, $d = 0.18$, 95% CI = [0.039 0.137]).

Comparing the correlation of representations across learning can identify asymmetry of change for paired words but does not provide information about *how* the structure of the pair changes. For instance, our previous analyses showed that GENDER changes relatively more than its target FEMALE, but it cannot tell us whether GENDER becomes more similar to FEMALE, or whether the changes are unrelated to its to-be-learned target; Fig. 4B. To investigate this, we calculated a single asymmetry measure by subtracting the correlation of the cue after learning and target before learning from the correlation of the cue before learning and target after learning to determine whether how the representations change relative to each other. Here, a positive value would suggest the representation of the target is drawn towards that of the cue, a negative value would suggest the cue is drawn towards the target, and a value of zero would suggest that the relative representational change of the cue and target is symmetric. An LMM on our

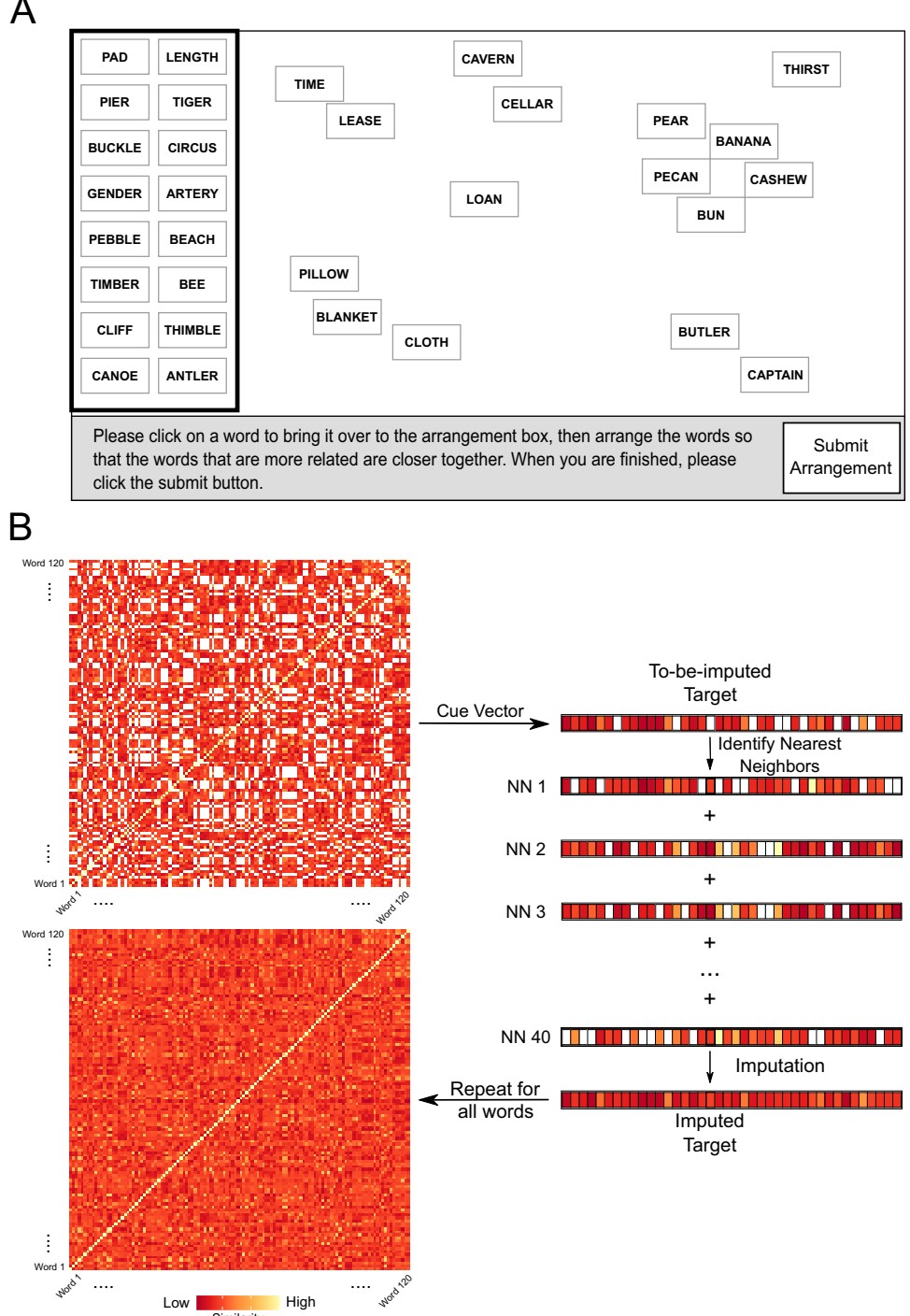

**Fig. 3 | Similarity-based word arrangement task (SWAT) and behavioral representational similarity analysis imputation approach. A** On each SWAT trial, a set of 60 words initially appeared in a random order in a box on the left side of the screen. Words would move to the main canvas when clicked, and once there could be dragged to a chosen location. Participants were instructed to place words that were more similar closer together but not given any rules for how to judge similarity. Participants could move words around until they were satisfied with their final arrangement. Words on the canvas in this figure are enlarged for readability. The assessment included four SWAT trials, and each word occurred on two of these trials. **B** Crucially, words from to-be-learned pairs never co-occurred on any SWAT trial to avoid potential contamination of their perceived relatedness, so the similarity of these pairs was imputed (see Methods). Euclidean distance was calculated for each pair of words as a proxy for dissimilarity and later converted to similarity for ease of interpretation. Lighter colors reflect pairs that are closer in semantic space, darker colors reflect pairs that were further away in semantic space. Note that the vectors displayed in this figure only show 40 values; the true imputation process would include all 120 potential similarity values.

asymmetry measure with fixed effects of relatedness, learning condition, and final recall success and a random effect of learning condition showed a significant main effect of relatedness ($t_{(452)} = 2.414$, $p = 0.016$, $\eta_p^2 = 0.01$, 95% CI = [0.006 0.064]), where the asymmetry value for related pairs was significantly different (more negative) from the asymmetry value for unrelated pairs; Fig. 6B. We additionally found that unrelated pairs did not show any significant asymmetry relative to zero ($t_{(78)} = 0.814$, $p = 0.418$, $d = 0.18$, 95% CI = [−0.023 0.055]). In

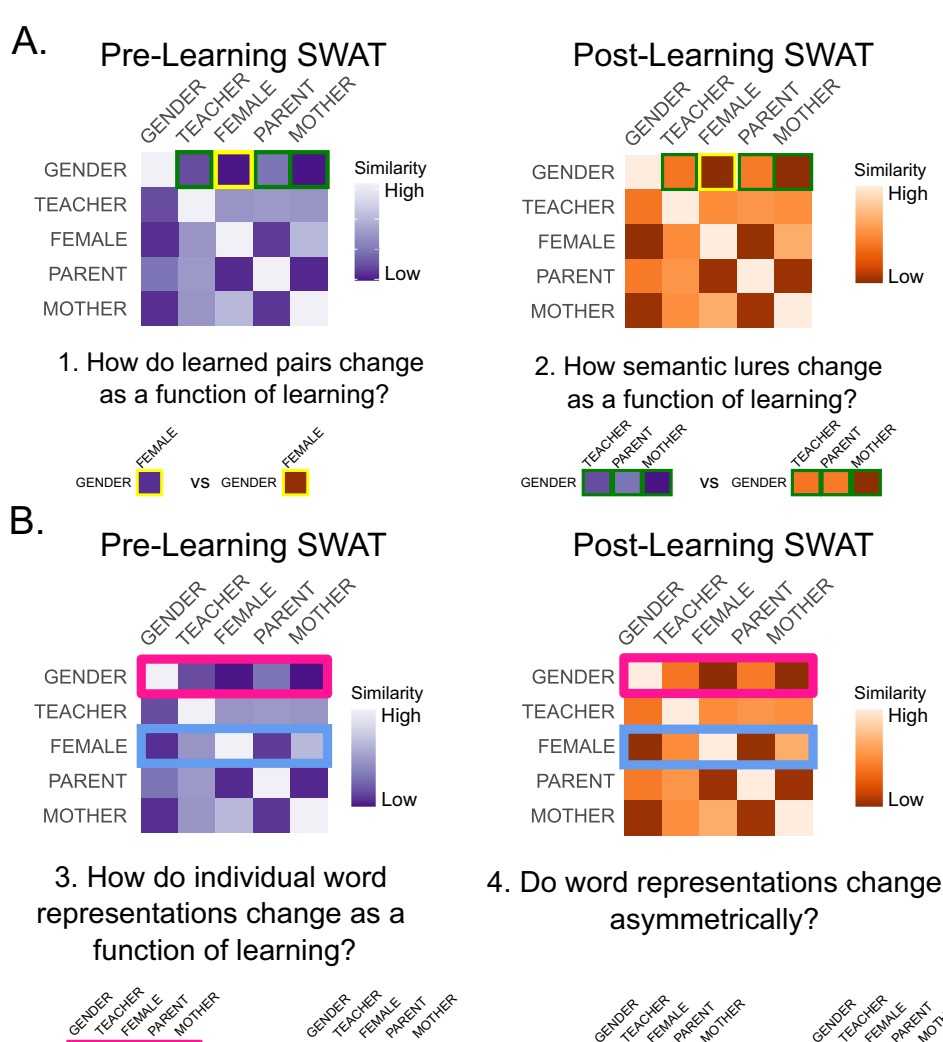

**Fig. 4 | Analyses of representational change. A** Schematic of representational similarity matrices (RSMs) derived from the Similarity-Based Word Arrangement Task (SWAT) procedure before learning (purple RSM) and after learning (orange RSM); note that our real matrices would be 60×60 words rather than the 5 × 5 words used in this toy example. Using the pair GENDER-FEMALE for illustrative purposes, we illustrate four of our key analyses. A. Analyses of pairwise representational changes across learning. In Analysis 1, cells outlined in yellow highlight the pairwise distance of the cue word GENDER to its target FEMALE, and we compare how this distance changes across learning. In Analysis 2, we examine the change in pairwise distance across learning between cue words (e.g., GENDER) and semantically related non-target words (lures; green outlines). **B** Analyses of individual word representations across learning. Pink outline reflects cue word GENDER, blue outline reflects target word FEMALE. We define the representation of an individual word as its row vector from the RSM (i.e. by its pairwise relationships to all other words in our set). In Analysis 3, we test how the representation of each word changes across learning by taking the Pearson correlation of the row vectors from the pre- and post-learning RSMs. In Analysis 4, we test whether the word representations in the to-be-learned pair change asymmetrically. In this analysis, we correlate the representation of the cue word before learning with that of the target word after learning and the representation of the cue word after learning with that of the target word before learning. The difference between these two values is calculated as a measure of asymmetry, where a positive value reflects the target being drawn towards the cue, a negative value reflects the cue being drawn towards the target and a value of zero reflects the cue and target being drawn towards each other symmetrically (or no representational change).

contrast, related pairs showed a numerically negative asymmetry value; however, despite a moderate effect size, this effect was only a nonsignificant trend after corrections for multiple comparisons ($t_{(79)} = 2.157$, $p = 0.068$, $d = 0.49$, 95% CI = [−0.049 −0.001]).

**Relating Recall Accuracy to Representational Change**
To further probe the behavioral relevance of representational change for learning outcomes, we conducted an item analysis (with each word pair considered an 'item'). The average accuracy at final test across

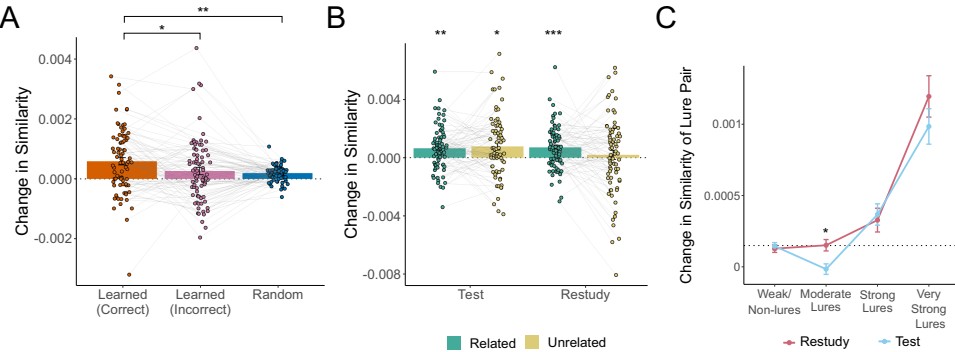

**Fig. 5 | Learning-induced changes in representational similarity for paired words. A** Linear mixed effects modeling revealed that pairs of words that were experienced as to-be-learned pairs (regardless of whether tested or restudied) and that were correctly recalled at final test (regardless of learning manipulation; orange bar) became more similar after learning than pairs that were not successfully recalled at final test (pink bar; $t_{(158)} = 2.566$, $p = 0.023$, $d = 0.20$, 95% CI = [$1.84 \times 10^{-5}$ $6.29 \times 10^{-4}$]) and arbitrary pairings of words that were never experienced as to-be-learned pairs (blue bar; $t_{(158)} = 3.112$, $p = 0.007$, $d = 0.25$, 95% CI = [$8.73 \times 10^{-5}$ $6.98 \times 10^{-4}$]). Y-axis indicates change in similarity (post-learning assessment minus pre-learning assessment). **B** Two-tailed one-sample t-tests reveal significant learning-induced similarity change for pairs of words that were tested (regardless of semantic relatedness; related: $t_{(79)} = 3.788$, $p = 0.002$, $d = 0.423$, 95% CI = [$3.045 \times 10^{-4}$ $9.804 \times 10^{-4}$]; unrelated: $t_{(74)} = 3.085$, $p = 0.017$, $d = 0.356$, 95% CI = [$2.736 \times 10^{-4}$ $1.272 \times 10^{-3}$]) and for semantically related restudied pairs ($t_{(79)} = 4.258$, $p < 0.001$, $d = 0.476$, 95% CI = [$3.763 \times 10^{-4}$ $1.037 \times 10^{-3}$]). Green indicates semantically related pairs, yellow indicates semantically unrelated pairs. For both panels A and B, open circles reflect means of individual participants ($N = 80$), with connecting lines showing within-subject differences across conditions. **C** Words that may interfere with successful recall of to-be-learned pairs (i.e. lures) were defined by their LSA cosine similarity to a given to-be-learned cue. Change in

similarity of these potential lure pairs was calculated across learning and entered into a linear mixed effects model based on data from 248,408 pairs of words across 80 participants. This model revealed a significant learning condition by lure strength interaction ($t_{(248400)} = 2.840$, $p = 0.005$, $\eta_p^2 = 3.25 \times 10^{-6}$, 95% CI = [$-3.130 \times 10^{-4}$ $-5.740 \times 10^{-5}$]), where very strong lures are drawn towards a given cue word regardless of learning condition (restudy: $z = 6.689$, $p < 0.001$, $d = 0.15$, 95% CI = [$6.59 \times 10^{-4}$ $1.48 \times 10^{-3}$]; test: $z = 6.029$, $p < 0.001$, $d = 0.12$, 95% CI = [$4.90 \times 10^{-4}$ $1.22 \times 10^{-3}$]), while strong lures showed no statistically significant change relative to weak/non-lures (restudy: $z = 1.707$, $p = 0.32$, $d = 0.022$, 95% CI = [$-3.94 \times 10^{-4}$ $7.93 \times 10^{-5}$]; test: $z = 1.964$, $p = 0.200$, $d = 0.022$, 95% CI = [$-3.76 \times 10^{-4}$ $5.02 \times 10^{-5}$]). In contrast, moderate lures are pushed away from cue words (relative to baseline weak/non-lures) more when the associated to-be-learned pairs are tested than restudied ($z = 2.840$, $p = 0.014$, $d = 0.03$, 95% CI = [$5.740 \times 10^{-5}$ $3.129 \times 10^{-4}$]). Maroon lines reflect lures associated with restudied pairs, blue lines reflect lures associated with tested pairs. Dotted line reflects average change in similarity for baseline weak/non-lure pairs. Across all panels, error bars reflect standard error of the mean. Symbols reflect statistically significant differences using two-tailed tests across conditions (panels A and C) or versus zero (panel B) using Holm-Bonferroni corrections for multiple comparisons (*$p < 0.05$, **$p < 0.01$, ***$p < 0.001$).

participants for each word pair (regardless of learning condition or semantic relatedness) was significantly correlated with its similarity after learning (Fig. 7A; $r_{(58)} = 0.46$, $p < 0.001$, 95% CI = [0.232 0.638]) and average change in similarity (Fig. 7B; $r_{(58)} = 0.39$, $p = 0.002$, 95% CI = [0.154 0.583]), suggesting that word pairs that are considered more similar after learning and that show greater learning-induced representational change are more likely to be remembered. Despite the relationship between pairwise change in similarity and behavioral accuracy, there was no statistically significant correlation between the magnitude of an individual's behavioral testing effect and their average change in similarity for tested and restudied pairs when averaging across words ($r_{(78)} = -0.163$, $p = 0.149$, 95% CI = [−0.369 0.059]).

Although comparing the average pairwise change in similarity to average accuracy across participants provides a valuable link between the re-sculpting of semantic space and behavioral performance, it overlooks the fact that semantic relatedness and learning condition may have differential effects on the recall success of a word pair. To investigate the relative contribution of these processes to behavioral performance, we conducted a mixed effects logistic regression predicting the Day 2 recall outcome of each individual word pair. Echoing our previous analyses, this model showed a significant relatedness by learning condition interaction ($z = 2.424$, $p = 0.014$, $\eta_p^2 = 0.0016$, 95% CI = [0.124 1.109]), in addition to significant main effects of relatedness ($z = 10.572$, $p < 0.001$, $\eta_p^2 = 0.028$, 95% CI = [−2.046 −1.406]) and learning condition ($z = 6.324$, $p < 0.001$, $\eta_p^2 = 0.010$, 95% CI = [0.815 1.547]). Follow-up tests revealed that there was a larger benefit of testing over restudying pairs on the probability of successful recall at Day 2 for unrelated pairs ($z = 12.431$, $p < 0.001$, $d = 0.20$, 95% CI = [−1.96 −1.43]) than related pairs ($z = 9.706$, $p < 0.001$, $d = 0.16$, 95% CI = [−1.55 −1.03]).

Additionally, this model revealed a significant main effect of the change in similarity of the cue representation ($z = 2.453$, $p = 0.014$, $\eta_p^2 = 0.0015$, 95% CI = [−0.772 −0.086]), suggesting that more change in the representation of the cue across learning is associated with a higher probability of recall at Day 2; Fig. 7C. Finally, this model showed a significant relatedness by condition by change in target representation across learning interaction ($z = 2.075$, $p = 0.038$, $\eta_p^2 = 0.0011$, 95% CI = [−1.678 −0.048]). Investigation of the slopes revealed that for tested unrelated pairs, there was a significant negative relationship between the probability of final recall success and the change of the representation of the target across learning ($z = 2.691$, $p = 0.007$, $d = 0.16$, 95% CI = [−0.266 −0.042]), suggesting that more change in the target across learning (i.e. lower correlation values) is associated with higher probability of subsequent recall. This slope was significantly more negative than the slope from unrelated restudied pairs ($z = 2.321$, $p = 0.020$, $d = 0.11$, 95% CI = [0.023 0.275]); Fig. 7D. There was no statistically significant relationship between change in target representation and probability of subsequent recall across learning for related pairs ($p$ values > 0.1; Supplementary Table 5). Together, these results show while the magnitude of representational change that a pair undergoes is associated with its probability of subsequent recall, there may be multiple processes underlying the change that depend on both the characteristics of the word pair itself and the learning conditions, and that these processes do not all impact the probability of successful recall.

## Discussion

Three primary questions were addressed in the current work. First, we sought to determine how semantic relatedness between paired words influences the testing effect. Second, we created an extension of a

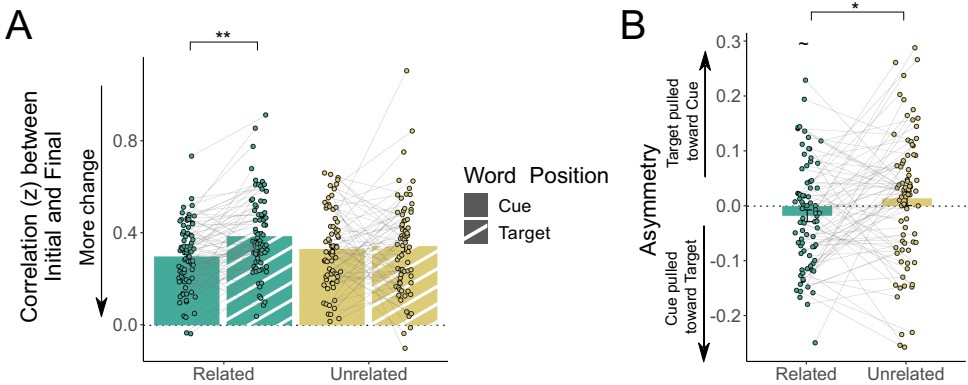

**Fig. 6 | Symmetry of representational change within pairs across learning.**
**A** Linear mixed effects modeling revealed that target words in semantically related pairs showed less representational change (i.e. a higher correlation between their initial representation and final representation) across learning than did cue words when correctly recalled at Day 2 ($t_{(372)} = 3.546$, $p = 0.002$, $d = 0.18$, 95% CI = [0.039 0.137]). This effect was not observed when comparing cues and targets in unrelated pairs. Green indicates semantically related pairs, yellow indicates unrelated pairs; solid bars indicate cue words, striped bars indicate target words. **B** Related pairs of

words show a significantly different (more negative) asymmetry in representational change than unrelated pairs ($t_{(452)} = 2.414$, $p = 0.016$, $\eta_p^2 = 0.01$, 95% CI = [0.006 0.064]). All displayed correlation values are Fisher $r$-to-$z$ transformed. Open circles reflect means of individual participants ($N = 80$), with connecting lines showing within-subject differences across conditions. Error bars reflect standard error of the mean. Symbols reflect statistically significant differences using two-tailed tests across conditions using Holm-Bonferroni corrections for multiple comparisons ($\sim p < 0.10$, $*p < 0.05$, $**p < 0.01$, $***p < 0.001$).

multi-arrangement similarity paradigm[49] to investigate how paired associate learning, supported by either testing or restudying, can shape the semantic representations of individual words. Finally, we assessed whether learning-induced changes in semantic representation were associated with behavioral performance.

To evaluate our first question, we systematically manipulated semantic relatedness between the cue and target with a to-be-learned pair of words and compared accuracy between tested and restudied pairs after approximately 24 h. We found that although relatedness increases overall performance, it decreases the magnitude of the testing effect by substantially improving performance for restudied pairs, such that the relative additional benefit conferred by testing is less than for unrelated pairs. Crucially, we only observed this interaction between semantic relatedness and learning condition when tested items were split between those successfully and unsuccessfully recalled at the initial testing. This is consistent with previous work[29,50,51] showing that, in the absence of feedback, the mnemonic benefits of testing only occur if the target item is successfully recalled during the initial test.

Accuracy alone, however, can only provide limited insight into exactly how semantic relatedness differentially improves memory for tested and restudied pairs of words. To address this gap, we developed an extension of a multi-arrangement paradigm to simultaneously measure the semantic similarity of sixty words at a time and impute the semantic similarity of words in to-be-learned pairs without them ever being directly measured against one another. In this analysis, we showed that successful learning, especially of related pairs, draws paired words closer together in semantic space more than unsuccessful learning attempts and pairs that did not undergo learning.

Showing that pairs are drawn together, however, does not show how they become more similar. It is possible that both items within a pair change symmetrically to become more similar to each other; alternatively, one item may remain relatively stable while the other changes. Extant literature investigating these potential hypotheses[39,40,52–54] tends to compare outcome measures like accuracy and reaction times when probing pairs in the forward (i.e. A→B) vs backward (i.e. B→A) directions. These measures, while useful for answering some questions, are less effective for exploring associative asymmetry of changes in semantic space, as they cannot compare the overall representations of concepts.

To this end, we compared the semantic representations of individual words across learning. We found that for related pairs, learning-induced greater representational change in the semantic structure of cue words than target words, while there was no statistically significant difference in the change in unrelated cue and target words. We then adapted an approach from neuroimaging literature for investigating asymmetrical representational change[41,55]. If the correlation between pre-learning cues with post-learning is less than the correlation between post-learning cues with pre-learning targets, this implies that learning draws cues towards targets in semantic space. This was indeed the pattern we observed for related word pairs (although as an isolated effect, the negative asymmetry value narrowly failed to survive corrections for multiple comparisons; however, the change in asymmetry relative to unrelated pairs was significant).

The idea that testing creates a directionally-specific (i.e. asymmetric) associative relationship, where the cue-to-target relationship is strengthened without influencing the backward associative target-to-cue link, is consistent with prior theoretical accounts. According to the dual memory theory[56] this process occurs by creating an episodic "cue memory" where the cue and target are encoded in the context of a retrieval task, whereas restudying creates a bidirectional association. The transfer-appropriate processing account[32] posits that the benefit of testing stems from greater episodic contextual similarity between retrieval practice and the final test, relative to restudying.

Consistent with this framework, our results show asymmetric change in cue and target representations across learning; however, this asymmetric change depends on the pre-existing semantic relatedness, rather than learning condition, suggesting that the asymmetrical change within a pair may be driven by the semantic information within the to-be-learned pairs, rather than the creation of an episodic "cue memory" during testing. Other work has suggested that prior knowledge plays a crucial role in the symmetry of concept representations after learning[39,41,52]. For instance, when pairs of famous and novel faces are learned, multivariate neural representations of novel target faces are drawn towards those of their paired cue faces only when there is pre-existing knowledge about the cue face[41]. While this asymmetric representation is in the opposite direction to the one we observed in our data, it is important to note that in that study there was no pre-existing relationship between the paired faces and no prior knowledge surrounding the novel faces. In contrast, the word stimuli used in our study had a rich network of semantic associations prior to

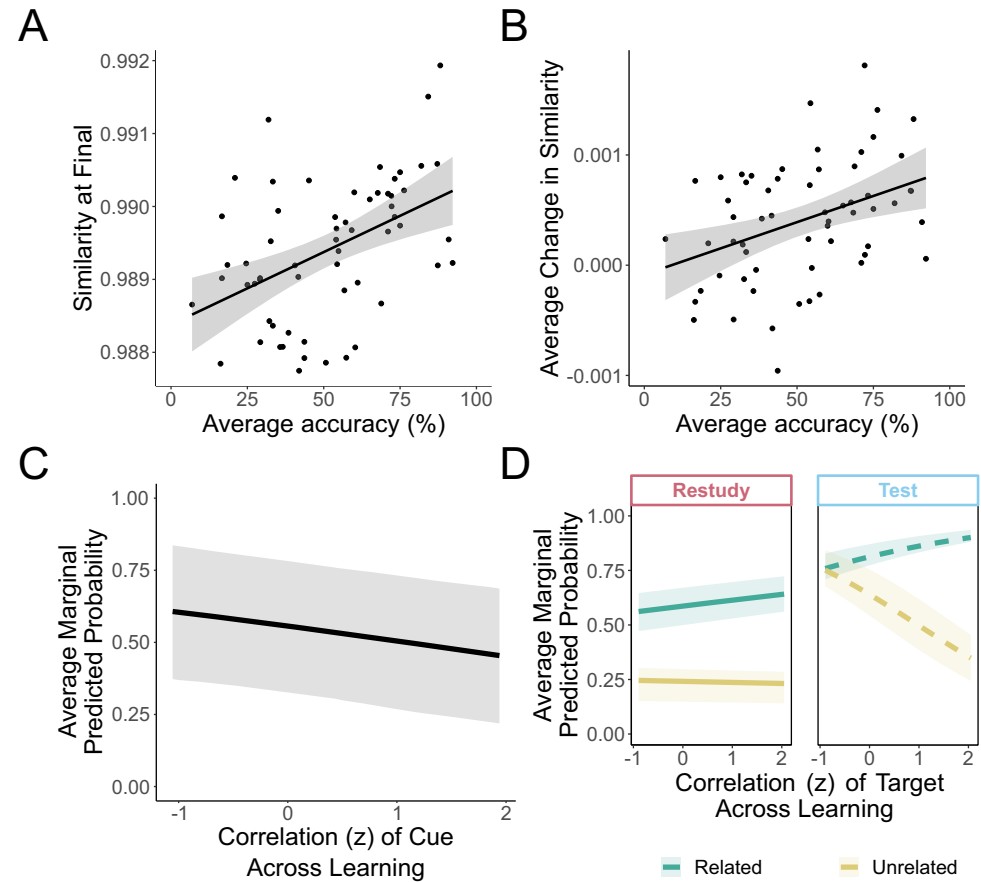

**Fig. 7 | Relating representational change to recall accuracy on the final test.**
**A/B**. When averaging across 80 participants, word pairs that showed greater within-pair similarity after learning (**A**; $r_{(58)} = 0.46$, $p < 0.001$, 95% CI = [0.232 0.638]), and those that showed more representational change (difference in within-pair similarity between final and initial assessment) (**B**; $r_{(58)} = 0.39$, $p = 0.002$, 95% CI = [0.154 0.583]) were recalled with greater accuracy at the final test. $N = 60$ word pairs. Closed circles reflect average similarity (**A**) or change in similarity (**B**) and average final test accuracy across 80 participants. Shaded area reflects 95% CI. **C** More change in cue representation after learning (i.e. lower correlation between initial and final) is associated with a higher probability of recall at Day 2 ($z = 2.453$, $p = 0.014$, $\eta_p^2 = 0.0015$, 95% CI = [−0.772 −0.086]). **D** The impact of the change in target representation across learning on subsequent recall depended on the semantic relatedness of the pair and whether the pair was tested or restudied at Day 1. For unrelated pairs that were tested at Day 1, more change in the representation

of the target across learning (i.e. lower values on x-axis) was associated with higher probability of subsequent recall ($z = 2.691$, $p = 0.007$, $d = 0.16$, 95% CI = [−1.678 −0.048]). No other relationships between target word representational change and behavior were significant. Green indicates semantically related pairs, yellow indicates semantically unrelated pairs. Dashed lines indicate tested pairs of words, non-dashed lines indicate restudied pairs of words. For both **C** and **D**, $N = 3902$ pairs of words from 80 participants. Values on x-axis reflect Fisher's z-transformed correlation of cue representations across learning (**C**) and of targets across learning (**D**); values on y-axis reflect average marginal predicted probability (probability of subsequent recall across all participants across change of the predictor of interest, holding other predictors constant). Shaded areas reflect 25th–75th percentile of the average marginal predicted probability. All tests were two-tailed and no corrections were made for multiple comparisons.

learning, with pre-existing semantic relationships between half of the pairs. It is possible that the assimilation of a target item representation into that of its paired cue item only occurs when existing semantic information about the cue can scaffold the integration of the novel information into the existing knowledge. When there is pre-existing knowledge about both items in a pair, as was the case in our study, the cue representation instead changes asymmetrically to become more predictive of the upcoming target[55,57].

Although words in our corpus that are strongly associated with a given cue word are drawn towards that cue word regardless of the learning condition or relatedness of the associated to-be-learned pair, we only show learning-induced asymmetric sculpting of the overall semantic space for semantically related pairs. Accounts of the testing effect such as the elaborative encoding account[58] or the semantic mediator hypothesis[31,33,59] propose that mental elaboration during the search for correct answer during testing facilitates later recall by dynamically creating additional retrieval routes via the activation of concepts connecting cues and targets[33,59] or by increasing relational

processing relative to restudying[38,60]. Other work has shown that pre-existing semantic relationships between words facilitate integration of the pair, potentially amplifying these effects[61]. It is possible that even though semantic associates of the cues in unrelated pairs create new links within the to-be-learned pair, the paired words are less likely to co-activate shared concepts enough to change the overall semantic space[42].

We additionally conducted a set of analyses comparing participants' idiosyncratic semantic representations (derived from the SWAT) to normative semantic representations (derived from word2vec) to test whether these elaborative connections made during learning are truly novel or reflect the sculpting of existing features; see Supplementary Note 4 and Supplementary Fig. 6. We show that after learning, words in tested pairs are drawn closer to their normative representations, suggesting that even though learning drives novel connections, testing shapes features that already exist, rather than adding entirely new features to a representation.

The non-monotonic plasticity hypothesis (NMPH) may also help explain the effects we observed of testing on representational change. This account posits that while testing and restudying both strongly co-activate representations of the paired items, testing additionally requires a search process for the to-be-learned target that induces moderate co-activation of other similar items (as has been previously shown to occur during the retrieval of highly similar episodic memories[62,63]), weakening these connections and reducing interference[42]. This is precisely what we found in our analyses of lure representations – testing exerted the biggest impact on moderate-strength lures, which were significantly repelled away from cue words, relative to weak/non-lure pairings. This effect is not only consistent with the predictions of NMPH, but also with an emerging body of work showing that competition adaptively distorts and repels overlapping episodic representations so they become less similar[46,64,65].

Our last goal was to evaluate the linkage between learning-induced changes in semantic representations and final recall success. To do so, we examined how the mean retrieval success of each word pair (averaged across participants) relates to its mean learning-induced change in representational similarity, and how individual differences in multiple factors affecting representational change relate to the subsequent recall of a given pair. Using the first approach, we showed that word pairs that undergo a greater amount of pairwise representational change (regardless of learning condition) are more likely to be remembered at the final test. Our individual differences approach showed that pairs are more likely to be recalled after a delay when the representation of the cue changes more across learning, while learning-induced change in the representational space of the target is only associated with final recall success in unrelated pairs that underwent testing. These findings highlight how changes in the representation of the cue (to make it more predictive of the target) are crucial regardless of learning condition, but it may only be necessary to sculpt the representation of the target to create elaborative links between words in a pair if they do not already exist.

One potential limitation of our work comes from our use of pairwise similarity metrics derived indirectly via imputation. If our imputation method was unreliable, it might cast doubt on our behavioral representational change results. We believe that this is not the case and have performed extensive validation analyses of our imputations (see Supplementary Note 2). We believe that our ability to impute the subjective semantic relatedness of pairs without ever having participants directly judge them is a key innovation of our work over existing approaches such as semantic priming and free association that can only show the relative magnitude of the effects of semantic relatedness through measures like accuracy and reaction time. Moreover, we expect our imputation approach will allow researchers to infer pairwise relationships without running the risk of biasing participants by presenting to-be-learned pairs before learning, nor evoking demand characteristics by having participants explicitly judge the similarity of already-learned pairs, which may occur in traditional multi-arrangement paradigms.

Another potential limitation could come from our admittedly restricted assay of semantic space. Due to experimental time constraints, we were unable to include additional words beyond those in the to-be-learned pairs in our SWAT protocol that would enrich our measurement of semantic space and serve as a null hypothesis test, as they should undergo little or no representational change. We also ensured that the distributions of semantic association across conditions did not overlap so that we could treat relatedness as a dichotomous variable and actively avoided very strongly related pairs of words so that participants could not easily guess the target word in the absence of successful learning. These design constraints may have resulted in a truncated range of semantic relatedness across all pairs. Recent work has shown that the effect of semantic relatedness may depend on the range of strength of association across the entire

stimulus set[15], so future work may opt to choose a broader range to determine if this impacts the results.

Despite the general stability of semantic knowledge over the course of one's lifetime, our results demonstrate that even a brief session of episodic learning can subtly yet systematically re-sculpt semantic space. Our behavioral representational similarity approach identifies multiple processes supporting episodic memory, where new connections are established between a cue and target, shared semantic information asymmetrically changes cues to become more predictive of their paired target and testing minimizes associations with potentially interfering semantic lures. Together, these changes impart a lingering residue on semantic memory that facilitates later episodic recall. These results are consistent with recent neuropsychological, behavioral, and neuroimaging evidence that the episodic and semantic memory systems may interact through gradients of activation of shared cognitive processes[5–7]. In this framework, episodes are comprised of both general conceptual reinstatement and episode-specific sensory processing, while recall of semantic memory often includes episodic information about when and where the information was acquired[3]. Future studies will be needed to better characterize whether these subtle learning-induced semantic distortions are short-lived or whether they can endure for weeks or months.

## Methods

The experimental design and data analysis plan were preregistered prior to data collection on November 19th, 2020 on the Open Science Framework at https://osf.io/5q6th/.

### Participants

Participants were recruited via Prolific (https://www.prolific.co/) and through the UCLA SONA Undergraduate Participant Pool. A power analysis (see Supplementary Methods for details) suggested we would need a sample size of at least 73, so we aimed to collect useable data from 80 participants. A total of 262 participants (145 from SONA, 117 from Prolific) completed the first session of the experiment. Of those, 183 returned for the second session within 28 h of completing the first (88 from SONA, 95 from Prolific). After excluding participants who did not complete both sessions or who otherwise did not meet our strict inclusion criteria (described in the Supplementary Methods), we were left with 29 from SONA and 51 from Prolific. Participants from Prolific received monetary compensation and participants from SONA received course credit. The two samples were not significantly different on any key measures, so the samples were combined for a final $N = 80$ (29 male; age range = 18–39, mean age = 24.33, SD = 5.49). Participants from SONA had all completed at least high school level education; years of education was not collected from participants from Prolific. All participants provided informed consent prior to participating. This research was approved by the IRB of the University of California, Los Angeles. Participants from the UCLA SONA Undergraduate Participant Pool were compensated with course credit and participants recruited on Prolific were compensated at a rate of $7.00/hour.

Additionally, we noted in our pre-registration that we would exclude participants who reported rehearsing word pairs between sessions. Ultimately, we included the 8 participants who reported rehearsing word pairs between sessions, as we did not explicitly instruct participants not to rehearse and our survey question was not specific enough to determine the extent to which they rehearsed (i.e. it did not distinguish whether they spent hours rehearsing all word pairs, or just happened to spontaneously recall one or two of them).

### Material

Stimulus materials included 60 cue-target word pairs. Thirty of these pairs were semantically related and were drawn from the FSU Free Association Norms[66]. We restricted words to nouns with no

homographs, a concreteness norm greater than 3.5, and deemed by Nelson et al. as appropriate for use in an experiment because they had of an acceptable number of normed associates. In order to reduce the possibility that a participant might simply guess the target word given the cue word, pairs were restricted to have a forward strength of association less than 0.5, meaning that fewer than half of people who saw a given cue word would generate the target word in a free association task. Finally, any pairs of words that together made a compound word or were similar to any English idiom were excluded.

For each of the related pairs, we compiled three measures of pair similarity: (1) forward association strength, (2) cosine similarity from latent semantic analysis (LSA) derived from a corpus of 100k English words (http://www.lingexp.uni-tuebingen.de/z2/LSAspaces/) and the *LSAfun* R package[66,67], and (3) word2vec similarity, based off of a model trained on a subset of the Google News dataset, which contains 300-dimension vectors for 3 million words and phrases (https://code.google.com/archive/p/word2vec/). An additional 30 low relatedness pairs were selected to form the remaining 30 unrelated pairs. Target words of these pairs were shuffled until all 30 pairs had word2vec, LSA cosine similarities and (if the pair was normed), cue-to-target association strengths that were lower than the entire list of related word pairs to ensure no overlapping measures.

## Procedure overview

Participation in this experiment took place over two days, with the sessions occurring no more than 28 h apart (see Fig. 1 for a schematic of the procedure). On Day 1, participants first performed a multi-dimensional similarity rating task using a drag-and-drop interface (Fig. 2A; similar to an approach from work in neuroimaging[49], which used picture stimuli instead of words). Following this similarity-based word arrangement task (hereafter referred to as the SWAT), participants completed a learning task, where they were given two opportunities to initially learn a set of 60 words pairs (30 related; 30 unrelated). We note that in our pre-registration of this experiment we had stated that participants would only have one initial learning opportunity before the test/restudy manipulation; however, pilot data suggested that one learning opportunity was not enough to yield sufficient accuracy on Day 2. Then, participants were given a third opportunity to engage with each pair via either testing or restudying. Last, participants completed a short questionnaire about how distracted they were during the task. Participants received a link to the second part of the experiment the following day; if they did not complete the Day 2 session within 28 h (i.e. before they had a second night of sleep), they were excluded from all analyses. The Day 2 session (Fig. 1) began with testing of all word pairs ("final test"), and then participants performed another set of similarity judgements using the SWAT protocol. Testing was performed prior to the SWAT protocol on Day 2 to prevent the possibility that words encountered during the SWAT trials would trigger additional retrieval practice or other rehearsal, which could have influenced final test performance in unpredictable ways.

## Word pair learning

Participants performed three rounds of word pair learning. During each of the first two rounds, all 60 pairs were presented on the screen in randomized order, with the text written in capital letters. Each pair appeared for 4 s with a 2 s ISI. For each pair, the cue word was presented on the left and the target word on the right. During the first round, participants were asked to make a judgement about how related the cue and the target word pairs were on a scale of 1–4, with 1 meaning "not related" and 4 meaning "very related". The second round was structured the same as the first, but participants were asked to judge how likely it would be for those two words to appear on the same page of a book or magazine on a scale of 1–4, with 1 meaning "not at all likely" and 4 meaning "very likely". These judgements allowed for

incidental encoding and encouraged the relational processing of the words in each pair. Relatedness judgements are described in Supplementary Fig. 1 and Supplementary Note 1.

In the final learning round, 30 of the pairs underwent retrieval practice (testing) and the other 30 were restudied. Participants were instructed that if they saw the cue and target words together (just as they had in the prior two rounds) their task was simply to type the target word into the answer box; if they saw the cue word accompanied by four question marks ("????") their task was to attempt to recall the target word and type it into the answer box. If they could not remember the target word, participants were encouraged to take a guess, or they could leave the box blank. Asking participants to type the paired words in the restudy condition, rather than having them make an additional relatedness judgement as in the first two learning rounds, allowed us to match the behavioral response with that of the testing condition (i.e. typing a word). This also served to reduce the differences between behavioral responses in the restudy condition and the final test, where all pairs would be probed by asking the participant to type a word. The learning condition manipulation was randomly interleaved; although this interleaved design necessitates task switching within the learning opportunity block, there was no statistically significant difference in final recall accuracy between trials where the participant switched between testing and restudying and those where learning condition was consistent across consecutive trials (see Supplementary Note 1 for more detail).

Participants were not given a time limit on recalling the second word in the pair. No feedback was provided, as feedback can provide an additional restudying opportunity that can enhance final test performance for tested items[68] and inflate testing effects[28].

The assignment of the word pairs to either the test or restudy condition was counterbalanced by creating two matched sets of pairs with 15 related and 15 unrelated pairs. Words were always presented in the forward order (i.e. cue was always presented before the target). The sets were matched on concreteness, frequency, length of cue and target, word2vec and LSA cosine similarity measures. Each set of words was randomly assigned to either the test or restudy condition independently for each participant. Memorability of the pairs of words was measured post-hoc by computing the average recall accuracy of the pair across participants; there was a range of accuracy across pairs, ranging from 91% (GENDER- FEMALE) to 5% (CHILDREN – BIRD) of participants recalling any given pair (Supplementary Fig. 2). Despite the range of memorability across all pairs, there was no statistically significant difference in mean memorability across the two sets of words pairs (see Supplementary Note 1 for more details).

## Final test

In the final test, performed on Day 2, participants were presented with cue words from pairs they had learned on the previous day (with the cue word on the left and "????" on the right, just as in the testing condition on Day 1) and were asked to type in the corresponding target word. There was no time limit on recall, and participants were encouraged to guess if they couldn't remember the pairs or otherwise leave the box blank. Responses were scored as correct if they were spelled correctly or if a spell-checking algorithm (https://textblob.readthedocs.io/en/dev/) identified the correct target word as the most likely word.

## Similarity-based word arrangement task (SWAT)

The SWAT was performed at the beginning of Day 1, prior to learning word pairs, and again at the end of Day 2, after the final test. Each session of the task was comprised of 4 trials. On each trial, participants received 60 words in a "word bank" on the left side of the screen. Participants clicked on a word to bring it over to a main arrangement area ("the canvas") and then dragged each word to the location of their choosing. Participants were instructed to take as long as they needed

to arrange the words such that more similar words were closer together and more dissimilar words were further apart. Trials lasted a median duration of 7.14 min.

Individual words were pseudo-randomly assigned to trials based on the to-be-learned pairs. The list of cues and targets were each split in half, to create 4 lists of 30 words. Each list was paired with each other list, except for the list that would form the to-be-learned pairs. This procedure created 4 trials of 60 words each, ensuring that each word would be arranged twice and that the two words in each to-be-learned pair were never both encountered on the same trial. This was an important constraint, as the mere act of thinking about the semantic relationship of the words in the to-be-learned pairs (or learned pairs in the case of the post-learning assessment) during a SWAT trial could bias participants' word placement decisions and corrupt our ability to sensitively measure the behavioral consequences of our experimental manipulations. The order of the 4 trials was randomized for each participant, as was the order of the words in the word bank on each trial.

### Derivation of semantic similarity metrics

After participants completed the SWAT arrangements, semantic dissimilarity was calculated for each pair of words by taking the Euclidean distance between the locations of each pair of words on the canvas (measured as the distance in pixels from the center of each word). Trials were combined using an evidence-weighted average of scaled-to-match distance matrices[49]. However, because words within to-be-learned pairs were never included on the same trials, we could not directly measure the distance between these words. Thus, by design, our procedure produced an incomplete representational dissimilarity matrix. In order to reconstruct one of our primary measures of interest (i.e. the semantic distance between words in to-be-learned pairs, both before learning and after learning), SWAT trials were combined using an evidence-weighted average and the semantic dissimilarity of unmeasured data pairs was imputed using K-nearest neighbors imputation using the KNNImputer function[69] from Python's *sci-kit learn* package[70] with 40 neighbors (as was determined as an optimal number of nearest neighbors for imputation in simulations) and the "distance" weighting function (see Fig. 3B for a visualization of this process). This imputation procedure was performed separately on each participant's pre-learning SWAT data and post-learning SWAT data. Since the imputation of not-directly-measured semantic distance ratings is a key innovation of our experimental paradigm, we conducted a number of analyses to confirm the validity of the imputation, and these are described in the Supplementary Note 2 and Supplementary Figs. 3, 4. Finally, semantic dissimilarity measures were converted to similarity measures for ease of interpretation by taking 1 – dissimilarity and the upper triangle of the fully imputed similarity matrix was used for further analysis. This process resulted in a range of similarity values from 0.9765 to 0.9963 for the pre-learning SWAT ($M = 0.9890$, SD = 0.0028) and 0.9733 to 0.9972 ($M = 0.9894$, SD = 0.0032) on the post-learning SWAT.

### Statistical analyses

All statistical analyses were conducted in R (version 4.1.2; R Core Team, 2021) and visualized using the *ggplot2* R package (Wickham, 2016). A list of packages used (including version information) is included in the Supplementary Methods. Data and code are available on OSF at https://osf.io/5q6th/. Tests of normality are not reported given that t-tests, ANOVAs and linear mixed models are generally robust to violations of normality, especially with larger sample sizes[71,72].

### Preregistered analyses

To investigate how semantic relatedness influences the testing effect, accuracy for tested and restudied pairs was calculated separately for semantically related and semantically unrelated pairs for each participant in the final session. A $2 \times 2$ (relatedness x learning condition) repeated measures ANOVA (RM-ANOVA) using the *rstatix* package[73] was performed to detect differences between conditions on the final test. Furthermore, a single measure of the testing effect on behavioral performance was calculated for each semantic relatedness condition (related pairs, unrelated pairs) and all pairs (regardless of condition) by taking the difference between the probability of a tested item being correctly recalled and the probability of a restudied item being correctly recalled. Next, tested pairs were split based on whether they were correctly recalled on Day 1. The accuracy on Day 2 was assessed in another $3 \times 2$ RM-ANOVA (Day 1 condition (correctly recalled, incorrectly recalled, restudied) x relatedness (related, unrelated). Although we initially preregistered that we would include all trials in the remainder of our analyses, we ultimately opted to exclude pairs that were tested and incorrectly recalled at Day 1 (mean number of pairs excluded=11.23, SD = 4.57) because we were primarily interested in the effects of successful testing compared to restudying. Effect sizes for RM-ANOVAs are reported using generalized eta-squared ($\eta_G^2$), which measures the effect size with variation from other effects and includes variance due to individual differences[74].

Change in semantic similarity was calculated for each word pair by taking the difference between similarity on Day 1 and Day 2. With this change measure, a negative value indicates that words within a pair became less similar over time (initial similarity > final similarity), while a positive value indicates that words within a pair became more similar over time (final similarity > initial similarity). As a manipulation check, the Day 2 semantic similarities of learned pairs (i.e. pairs of words that were either tested or restudied in the main part of our experiment), split by those that were correctly recalled at Day 2 and those that were not, were compared to random, unlearned pairs (i.e. all possible pairings of words from our stimulus set that were not restudied or tested during the learning portion of the experiment). This test provides a noise ceiling (as any changes in unlearned pairs can be thought of as noise) and ensures that learned pairs indeed show more representational change than unlearned pairs. Next, values for the learned pairs were entered into a linear mixed-effects model (LMM) with fixed effect predictors of semantic relatedness, learning condition, and final recall success, and a random intercept of subject identity. We note that although these analyses were initially preregistered to use RM-ANOVAs and paired t-tests, we report our results in an LMM framework to be consistent with our exploratory analyses (see below) and account for variance from potential random effects and report the results from the RM-ANOVA in the Supplementary Note 3. A testing effect measure for similarity was calculated in a comparable way as we did for the memory recall performance data, by taking the difference between the raw similarity on the final day and the change in similarity across days for tested items and restudied items. These measures of the testing effect from the similarity data were correlated with the testing effect measure for performance in the learning task across all pairs.

Additionally, we evaluated the asymmetrical representational change of each individual word by extracting the vector of similarity comparing each word to its top 20 nearest neighbors before and after learning. Analyses were restricted to the 20 nearest neighbors to reduce the influence of distant words in semantic space, which would be relatively uninformative for the definition of a given word. For example, it is much more useful to consider the definition of BLANKET in relation to words like PILLOW or CLOTH (the top two closest neighbors in our set, as defined by word2vec), where one can consider the specific connection or compare features, than its relationship to MATH or CHEF (the two least similar words in our set), where they share few features or associates. Nearest neighbors were identified by calculating the cosine similarity between the full semantic feature vectors extracted from word2vec and selecting the top 20 largest similarity values, excluding pairs of words where the distance is imputed. The similarity values for these pairs as measured by the

SWAT were used as the vectorized representation for each word. The representation of the cue in the initial pair was then correlated with the target in the final pair $r(Cue_{Day1}, Target_{Day2})$ and the cue in the final pair to the target in the initial pair $r(Cue_{Day2}, Target_{Day1})$. Taking the difference between the Fisher z-transformed correlation values $r(Cue_{Day1}, Target_{Day2}) - r(Cue_{Day2}, Target_{Day1})$ provided a single measure to index the amount of asymmetric change of each of the individual words, where a positive value would indicate that the target word becomes more similar to the cue word, while a negative value would indicate that the cue was drawn more towards the target, and a zero value would indicate that there was equal change for each word in the pair. Asymmetry values were Fisher z-transformed and entered into a linear mixed-effects model (LMM) with learning condition (tested vs restudied), position of word in pair, and relatedness of pair as fixed effect predictors and subject identity as a random intercept. Semantic relatedness and learning condition were iteratively tested as potential random slopes using likelihood ratio tests (using *varCompTest* from the *varTestnlme* R package[75]) and the variance of random effects in the final model was estimated using restricted maximum likelihood (REML). Follow up pairwise comparisons were used to investigate significant effects with Holm-Bonferroni corrections for multiple comparisons. Additionally, we tested whether the Fisher z-transformed asymmetry values were significantly different from zero using a series of two-tailed one-sample t-tests with Holm-Bonferroni corrections for multiple comparisons.

We note that we initially pre-registered that we would complete this analysis using all values from the row vector (rather than just the top 20 nearest neighbors). This analysis was initially attempted and resulted in no statistically significant results. However, this analysis assumes that the measured representation of each word in our set is independent from that of the other words in our set; in a neuroimaging-based representational similarity analysis (which our analysis was inspired by), this is indeed the case. However, in our paradigm, the semantic representation of each individual word is derived from its relationship to every other word in the set, and all of these words also underwent learning. As such, when comparing the representation of a given word across learning to all other words in the set, we are unable to isolate the change of that specific word from the changes in all the other words in the set, thereby inducing additional noise and making it more difficult to see any meaningful change for any individual word.

Finally, we note that we deviated from our pre-registration for both our analyses of the change in similarity and asymmetry values of learned pairs by separating pairs into those that were subsequently recalled at Day 2 and those that were forgotten at Day 2 (and include this distinction as a fixed effect predictor in our models) to allow us to test how our findings related to behavioral performance. Additionally, although we include observations about trials that were tested but incorrectly recalled at Day 1 in our basic behavioral analyses, we opted to exclude those trials from our analyses of representational space as we were primarily interested in the differential effects of our learning conditions (which theoretically only occur when testing is successful[50]), and we ultimately did not have a sufficient number of trials that were tested and incorrectly retrieved at Day 1 to sufficiently power any analysis of representational change for that trial type.

## Exploratory analyses

As a complement to the preregistered analyses described above, several exploratory analyses were also performed. First, we performed an additional two-tailed paired t-test on the recall accuracy of tested pairs at the initial Day 1 test to determine whether semantically related pairs were recalled better than semantically unrelated pairs. Restudied pairs were excluded from this analysis as the accuracy of these pairs reflected the ability to correctly type the fully visible target word, rather than memory recall performance.

In addition to conducting a $2 \times 2$ RM-ANOVA on the similarity measures, we conducted a series of two-tailed one sample t-tests with Holm-Bonferroni adjustments for multiple comparisons to test whether the change in similarity in each condition was different from zero.

To further probe the effects of learning on semantic representations and representational change, we performed a series of LMMs, using the *lmer* function from the *lmerTest* R package[76] to estimate fixed and random effects. For each model, we included predictors of relatedness (related vs unrelated), learning condition (tested vs restudied), and recall success at Day 2 (recalled vs forgotten). Additional predictors were included for some models as necessary. Subject identity was entered as a random intercept for each model (which allows for variance in the intercept over participants), and semantic relatedness, learning condition, and position in pair (when relevant to model) were tested as potential random slopes sequentially using likelihood ratio tests (using *varCompTest* from the *varTestnlme* R package[75]) for each model separately. Although the potential variance in the slopes was not the primary target of these analyses, the inclusion of random slopes allowed us to better explain variance in the model overall. All models were run with a maximum of 200,000 iterations for convergence. Once the final model was determined, significant main effects and interactions were probed using pairwise comparisons (using the *emmeans* R package[77]) with Holm-Bonferroni corrections for multiple comparisons. All models were estimated using REML, and two-tailed t-tests for fixed effects were estimated using Kenward-Roger's method. Effect sizes were estimated using partial eta-squared ($\eta_p^2$), as measured from the *effectsize* R package[78]. Unless otherwise noted, this procedure was used for all LMMs. Tables listing all coefficients, standard errors, degrees of freedom and t-values, in addition to variance-covariance structure for each model are reported in Supplementary Tables 3–15.

In addition to our preregistered analyses of representational asymmetry described above, which operate on pairwise similarity values of cues and targets before and after learning, we also sought to analyze how each word within a given pair underwent representational change. To test this, we first computed each word's similarity with its top 20 nearest neighbors, and thus derived a 20-value representational vector for each word before and after learning. We used the Fisher z-transformed Pearson correlation between these vectors as a measure of change for each individual word. In addition to the fixed effect predictors of relatedness, learning condition, and recall at Day 2, this model included a fixed effect predictor of the word's position in the to-be-learned pair (cue vs target). This effect of word position was also tested as a potential random effect using likelihood ratio tests, as was done in previous models.

We additionally explored changes in the semantic distance of potentially interfering lure pairs (i.e. words in our set that were semantically related to the cue words of our to-be-learned pairs) to further explore the sculpting of semantic space due to learning. To do so, we calculated the semantic similarity (indexed by the LSA cosine similarity) for all potential 118 pair combinations for a given cue word in our to-be-learned set of words (excluding the associated to-be-learned target word and a word's similarity to itself). Given that these pairs were identified post-hoc after creation of the to-be-learned pairs, there was a wide range of similarity values. We then divided these lures into four classes of lures: weak/non-lures (LSA cosine similarity less than 0.2), moderate lures (LSA cosine similarity between 0.2 and 0.4), strong lures (LSA cosine similarity between 0.4 and 0.6) and very strong lures (LSA cosine similarity above 0.6); Supplementary Fig. 5. For example, for the to-be-learned pair BLANKET – BED, SEMESTER would act as a weak/non-lure, TEMPERATURE would act as a moderate lure, SEAM might act as a strong lure and PILLOW would act as a very strong lure. Pairs that were not correctly recalled at Day 2 were excluded from this analysis, as incorrect responses were often other words from our corpus (which would be considered lures in this

analysis) and this retrieval may have influenced the similarity judgements in SWAT protocol (which was performed after the final test). Additionally, as in our other analyses, we excluded tested pairs that were incorrectly recalled at Day 1. This selection was repeated for the cues of all to-be-learned pairs separately for each individual, resulting in a range of 1652–5900 (mean=3106, SD = 981) semantic lure pairs per participant. We used pairwise change in similarity across learning for the semantic lures as the dependent variable for an LMM regression with fixed effects of condition of the associated to-be-learned pair (tested vs restudied), relatedness of the associated to-be-learned pair (related vs unrelated), and strength of the lure pair (weak/non-lure, moderate lure, strong lure and very strong lure). This model used a BOBYQA optimizer to ensure model convergence. As in our previous analyses, subject identity was included as a random effect in all models and relatedness and learning condition were independently and sequentially tested as potential random effects, as were potential two-way and three-way interactions. Significant main effects and interactions were probed by computing the contrast of the difference of each lure class and the non-lure pairs and comparing across learning condition (for example, the contrast [moderate lure – weak/non-lure for tested pairs] – [moderate lure – weak/non-lure for restudied pairs]) with Holm-Bonferroni corrections for multiple comparisons. Additionally, pairwise comparisons of all lure classes across learning condition were computed with Holm-Bonferroni corrections for multiple comparisons and are reported in the Supplementary Note 3.

To supplement our analyses relating representational change and semantic structure to final recall success, we ran a generalized LMM with a logit link function (i.e. a mixed-effects logistic regression) using the *glmer* function from the *lme4* package[79]. This model was fit using maximum likelihood estimation and a BOBYQA optimizer with a maximum of 200,000 iterations. We included fixed effects of learning condition (tested vs restudied), relatedness (related vs unrelated), Fisher z-transformed correlation of the cue and target across learning, difference of Fisher z-transformed correlation to normative semantic space across learning for both cues and targets, and Fisher z-transformed asymmetry value to predict the probability of final recall success (recalled vs forgotten). As in our previous LMMs, the effect of subject identity was included as a random effect, and random effects of relatedness and learning condition were independently tested as potential random effects using likelihood ratio tests. Significant main effects and interactions were probed using pairwise comparisons with Holm-Bonferroni corrections for multiple comparisons.

### Reporting summary

Further information on research design is available in the Nature Portfolio Reporting Summary linked to this article.

## Data availability

The raw behavioral data generated in this study have been deposited in the Open Science Framework at https://osf.io/5q6th/ (https://doi.org/10.17605/OSF.IO/5Q6TH). LSA cosine similarity data are available at http://www.lingexp.uni-tuebingen.de/z2/LSAspaces/. Pre-trained word2vec model is available at https://code.google.com/archive/p/word2vec/.

## Code availability

All code necessary to reproduce all analyses in this manuscript are provided at https://osf.io/5q6th/ (https://doi.org/10.17605/OSF.IO/5Q6TH).

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

## Acknowledgements

The authors would like to thank Craig Enders for his insight into how to handle missing data. This material is based upon work supported by the National Science Foundation Graduate Research Fellowship Program under Grant No. DGE-2034835 (CRW). Any opinions, findings, and conclusions or recommendations expressed in this material are those of the author(s) and do not necessarily reflect the views of the National Science Foundation.

## Author contributions

C.R.W. and J.R designed the experiment. C.R.W. collected and analyzed the data; J.R. provided guidance. C.R.W. wrote the initial draft of the paper; C.W. and J.R. revised and edited the paper. J.R. acquired funding.

## Competing interests

The authors declare no competing interests.
