## [Peer Review File · Nature Communications]

Behavioral representational similarity analysis reveals how episodic learning is influenced by and reshapes semantic memoryREVIEWER COMMENTS

Reviewer #1 (Remarks to the Author):

Review: Walsh & Rissman, 2023

Comments Summary

The authors report some exciting work on how semantic and episodic memory interact with each other, and with different approaches to learning (i.e., re-studying vs. testing) during paired-associate learning. It is encouraging to see work being done on the reciprocal influence of semantic and episodic memory, in addition to the application of interesting statistical approaches in the behavioural experiment space (e.g., representational similarity analysis). In reading the article, some notable questions remain about the overall design of the task which require clarification. There were also some cohesiveness issues in the article as a whole, and in particular sections that are outlined below. In the comments provided, I highlight these outstanding questions and offer some suggestions for improving the writing.

Major Comments

1. You highlight how semantic relatedness can promote memory in the face of interference, but it is also well-documented that semantic knowledge can cause interference, leading to poorer memory (e.g., proactive interference). Clarify what conditions would lead semantic memory to resolve, rather than cause interference, and explain what is meant by interference in this case (e.g., noisy environment? Competing memories? Etc.). This could also be elaborated on in the discussion.

2. More cohesion was needed in the introduction. It wasn't always clear how each section related to one another, or how they related to the main questions. I think the biggest problem was that the main questions were not introduced early enough. Including these in the first paragraph would be helpful to orient the reader and provide a better structure for the flow of the introduction. For example, the questions that are later outlined in the discussion, were not the main questions that I drew from the introduction. It would also be helpful to introduce the testing effect earlier as a topic of interest, as it was unclear why the authors jumped from talking about semantic relatedness to the testing effect.

4. The task needs more clarity in the introduction in order to set up the hypotheses and predictions. It would be helpful to clarify briefly what participants are doing, how the pairs were varied in terms of their semantic relatedness, and what was the difference between the two learning conditions. It was difficult to understand the predictions and keep track of the stimulus manipulations when they weren't well-explained in the introduction.

6. In the methods, the authors show that participants complete two study phases with relatedness judgments, followed by an intermixed restudy/test phase, then the final memory test following a delay. This design raises several questions about the mechanisms behind the findings. Firstly, the first two study phases are incidental learning tasks with a relatedness judgment. Thinking deeply about the semantic relationship between two words is a much different task than simply re-typing the word, as in the intermixed block. The differences between re-study vs. test could partially be attributed to shallow vs. deep processing of the pairs. While this is one of the arguments that is typically put forth when explaining the testing effect, the differences could be inflated by this manipulation. Please clarify why it was necessary to change the study method in the intermixed block, rather than having participants complete another relatedness judgment.

Additionally, there may be some effect of switching between a recall task and a typing task in the intermixed block. While this shouldn't have an impact on performance on day 1, given that participants had unlimited time to provide an answer, it could have impacted day 2, particularly for the re-studied items. This effect may occur because the re-tested items may be more difficult to

inhibit, particularly when the items are semantically related (again, shallow vs. deep processing). It may be worth including switching as a covariate in the model. In follow up studies, it may also be worth replicating the within-subjects effects, as well as running a between-subjects manipulation where the intermixed block is changed to either a re-study block or a test block depending on the condition.

Last, there is a literature on the benefit of one manipulation over another only when encoding or testing is on mixed list, as it was in this study, than on pure lists (e.g., on the superiority of intentional over incidental learning, in levels of processing paradigms (Craik, JEP :Gen, 2023) and on the memory superiority of emotional over neutral items (Talmi, et al., Journal of Memory and Language, 2007).

7. Please explain why the second SWAT task comes after the last memory test and not before. There are pros and cons in both situations, but there could be an additional boost from the testing effect that comes from the retrieval task.

8. Clarify whether the word pairs were always presented in the same order (e.g., did "road" always precede "car"?). While forward strength of association was reported, backward strength was not (i.e., how similar is the target to the cue). Would you expect similar representational changes in the second SWAT had the order of the pairs been reversed?

9. For recall accuracy it is worth noting how the results were potentially mediated by whether or not the word pairing was correctly recalled. This seems to also be specific to how testing effects and semantics interact with each other. Perhaps you could run something like a mediation analysis or include correctness as a covariate in an LMM.

10. Figure 5B is slightly confusing, perhaps relabelling the y-axis would be helpful. The way it reads now is that there is asymmetry for both related and unrelated pairs, but just in different ways. In the related pairs, the representation of the cue is moving towards the target, and in the unrelated pairs, the target is shown becoming more similar to the cue representation. You could relabel to something like the following:

0 to -0.3: targets ↓ cues; 0 to 0.3: targets ◊ ↓ cues

11. Lines 540-542: To better introduce the next section, briefly describe how these two mechanisms may interact.

12. Overall, each section of the results was straightforward, but it wasn't always clear why the next analysis was the logical next step. Similar to comment 11, you could better introduce the next section by highlighting a follow-up questions, for example.

13. As above, it wasn't clear whether the questions posed in the discussion were as explicit as they needed to be in the introduction. Making the changes to the introduction suggested above will improve the cohesion between the introduction and discussion.

14. In addition to the themes that have already been raised in the discussion, another relevant topic is the role of transfer appropriate processing in the testing effect (e.g., Morris, Bransford, & Franks, 1977). In the testing condition, you get the same task in the intermixed phase and test phase which promotes memory by increasing contextual similarity between encoding and retrieval. This is also relevant to the re-study phase, which is more discriminable from the retrieval context.

Minor Comments

1. Line 49: You state that early psychological theories proposed a separation between semantic and episodic memory systems. It is important to state whether this is a psychological distinction, neurological distinction, or both.

2. Lines 108-110: Redundant wording at the beginning of the two sentences, you can join them together.

E.g.,

"This theory proposes that changes in memory strengths are driven by the relative activation of items, such that memory for items that are strongly activated is strengthened, while items that are moderately activated are weakened or differentiated."

3. Evidence from the visual domain is brought up a few times in the introduction, and the way it is phrased makes it seem like a prominent aspect of the research being discussed. To minimize this confusion, you could de-emphasize the visual aspects in your phrasing. For example, you write: "In the visual domain, recent neuroimaging work has shown asymmetrical integration for novel faces paired with famous faces."

This could be rephrased as:

"Recent neuroimaging work has shown asymmetrical integration for novel faces paired with famous faces."

4. Lines 122-125: Delete "a" from the second last line of the sentence

i.e.,

"In the present study, we sought to directly investigate the influence of semantic relatedness on the testing effect and understand how episodic paired associate learning might sculpt pre-existing semantic space."

5. In setting up the demographics, please also report the average years of education + age range to get a better picture of the sample. The average years of education is of particular interest given its potential link to semantic ability (e.g., more flexible semantic networks).

6. In figure 1, it might be helpful to include a legend on the figure to illustrate which items were restudied and which were tested (i.e., red = restudied; blue = tested).

7. Lines 449-452: Delete "as" before the word "experienced."

i.e.,

"...showed greater change in similarity than did a control comparison of arbitrary pairings of words that were never experienced as to-be-learned pairs"

8. Line 610: Change the wording in this sentence. You say that you will address your first hypothesis, but you have not stated your hypotheses, you have stated your main questions.

9. Line 612: "...an approximately 24-hour delay" is a bit awkward, could reword to "after approximately 24 hours."

10. Line 636: Change "effects" to "affects."

11. Lines 664-668: this sentence is very long and does not read clearly, I suggest the following change:

"The Dual Memory Theory proposes that testing creates a directionally specific associative relationship, such that the cue-to-target relationship is strengthened without influencing the backward

associative linkage of the target to the cue. This is accomplished by creating an episodic “cue memory” where the cue and target are encoded in the context of a retrieval task, whereas restudying creates a bidirectional association.”

Reviewer #2 (Remarks to the Author):

This is an important contribution to the literature on the interplay between episodic and semantic memory, demonstrating how semantic representational space is potentially reshaped by episodic learning. Using a novel behavioural representational similarity analysis approach, the authors reveal differential reorganisation of semantic space based on the degree to which cue and target representations are related, with differences observed depending on the learning condition. These findings suggest that episodic learning might distort semantic space in an adaptive manner by strengthening associations within pairs and potentially reducing interference from lures.

Overall, I found this to be a very well-executed study. The authors are to be commended for developing an elegant behavioural paradigm and their meticulous attention to detail. The manuscript itself is extremely well-written; hypotheses are well articulated, and findings are interpreted in an appropriate and measured way. I have only very minor comments that the authors might like to address, on what is otherwise a fine piece of work.

Comments:

I wondered if there is a possibility that some of these words are more salient/memorable to participants over others? Does education play a role here or other individual differences? Did the authors collect any information regarding the time taken to organise the words during the SWAT task?

On page 8 the authors mention that word pairs that were incorrectly recalled at Day 1 were excluded from the analyses. How many of these pairs were excluded?

I would have liked to have seen further details regarding the exploratory analyses where semantic lures were identified based on their associated to-be-learned targets. How many pairs were tested and how were these proximities identified? It was not entirely clear to me how this was conducted.

The exploratory analyses, while very interesting, quickly becomes difficult to follow. I wonder if it is possible to include representative examples to help the reader follow exactly what is being tested in each section. By providing representative examples of semantically related and unrelated pairs, it may help to contextualise the research for readers who are not specialists in this field.

Minor typographical errors:

Figure 3 caption: “as a function semantic relatedness”
[“of” is missing]

Line 452: “never as experienced”
[remove “as”]

Line 458: “sematic”
[should read “semantic”]

Reviewer #3 (Remarks to the Author):

Summary

In this paper, Walsh and Rissman develop a novel spatial arrangement task (SWAT) to probe how learning word pairs of varying relatedness impacts the organization of participants' semantic knowledge, and how this learning is modulated by testing versus restudying the pairs. This task was developed to infer relatedness scores for pairs of words that were studied together. They find that related pairs of words benefit less from the testing effect relative to unrelated pairs (less of a difference in accuracy for tested versus restudied pairs), and using the SWAT, they find that word pairs are represented more similarly after learning, are repulsed from unlearned semantic neighbors (i.e. semantic lures), and that there are asymmetries in whether the target or cue words are impacted by learning. Finally, they find that semantic relatedness exerts opposing forces on how the learning-related changes in a target words' semantic representation relates to its later. The authors interpret their findings through the lens of the non-monotonic plasticity hypothesis and other computational frameworks.

Evaluation

Spatial arrangement tasks have long been used to infer semantic relatedness, but the imputation approach is an incredibly clever modification of the task that enables the authors assess the relatedness of word pairs without drawing attention to them or evoking retrieval strategies that may not necessarily reflect their semantic organization. I can imagine many other memory-related questions that would benefit from this task, and the many analyses that employ this task result in a rich dataset that can target different ways in which associations in memory can change. The question of the testing effect and how it is impacted by semantic similarity will be of interest to memory researchers as there has been a renewed focus on interactions between episodic and semantic memory in recent years. As you can see below, my main suggestions involve improving the clarity of the paper and expanding on the theoretical motivation and implications of the results.

Major comments

The major novelty of the paper is the development of the SWAT and the imputation approach that is used to characterize changes in how specific words were associated with each other and how their relationship to other words changed after learning. However, there is a long history of studying how recent/episodic experiences shape or update semantic knowledge that is not referenced here. Lots of examples can be found in the below papers for example. There are two topics that would strengthen the paper. The first is a discussion of the SWAT/imputation approach and its advantages/disadvantages relative to other measures of semantic relatedness such as semantic priming, free association, spatial arrangement without imputation, etc. The second is more of a theoretical discussion on the dynamic nature of semantic memory and how new learning can change and update semantic knowledge networks. For example, does exposure to GENDER – FEMALE decrease the path length between these two words, create a direct path that previously was only indirect, or expand the meaning of gender to include more 'female' features? Would testing or restudying affect the way that the semantic network is being changes? These are purely speculative questions but are fascinating to think about!

Yee, E., & Thompson-Schill, S. L. (2016). Putting concepts into context. *Psychonomic Bulletin & Review*, 23(4), 1015–1027. <https://doi.org/10.3758/s13423-015-0948-7>

Casasanto, D., & Lupyan, G. (2015). All Concepts are Ad Hoc Concepts. In E. Margolis & S. Laurence (Eds.), *The Conceptual Mind: New directions in the study of concepts* (pp. 543–566). MIT Press.

Connell, L., & Lynott, D. (2014). Principles of Representation: Why You Can't Represent the Same Concept Twice. *Topics in Cognitive Science*, 6(3), 390–406. <https://doi.org/10.1111/tops.12097>

Estes, Z. Attributive and relational processes in nominal combination. *Journal of Memory and Language* 48, 304-319, doi:10.1016/S0749-596X(02)00507-7 (2003).

Solomon, S. H. & Thompson-Schill, S. L. Finding features, figuratively. *Brain and Language* 174, 61-71 (2017)

The analyses reported are complex and there are times where a clearer description of the methods could greatly increase the readability of the Results section. Since each analysis is subtly different, it can be hard to keep track of the similarities and differences across the sections. It might be useful to start each section with an example – I found the example describing the semantic lures and to-be-learned pairs very helpful, and continuing that example throughout the paper might help to clarify and differentiate the many complex analyses that are reported here. For instance, the section on changes in semantic space could be well served by listing 5 of the 20 nearest neighbors that the vector for GENDER was compared to, etc. It might also be useful to have either a separate analysis approach figure showing what is being correlated when (cue to target, cue to target before learning, target to cue before learning, target to 20 neighbors, etc), or an equivalent schematic that visualizes and summarizes the main findings of the paper.

One drawback of excluding forgotten items from the retest condition on Day 1 is that the comparison to restudying is confounded, because items that are restudied on Day 1 will include a mixture of items that would have been remembered or forgotten if tested (both if tested in place of restudy, or if tested after a chance to restudy). This means that memory will pretty much guaranteed to be worse in the restudy condition relative to the test condition. I believe it's more common to compare all tested trials with all restudied trials and this is the approach that was pre-registered. This is a fairly large deviation from the pre-registration and it's worth mentioning it briefly in the Results section, not just in the discussion and supplement.

Figure 3: The A and B panels might be clearer if they were grouped by day instead of relatedness, with relatedness on the x-axis instead of day. Then the Day 2 panel would reflect all components of the 2x2 ANOVA that is the first analysis presented in the methods. It would also be a visual reminder that the Restudy data from Day 1 is never meaningfully compared to Day 2 since it is from a different task.

Word2vec model predicting the change in semantic relatedness: I personally don't see the value of including this in the main text unless there is a deeper discussion of how and why these results diverge from those observed with the SWAT. They may fit better in the SI. Regardless of where it ends up though, according to the Methods, this model included "the semantic relatedness of the pair, learning condition, recall success at final test and word's position in the to-be-learned pair" – but only the effect of learning condition is reported in the Results. All tested predictors should be reported here, even if combined, e.g. No other predictors were significant (all t's < X, a p's > 0.XX).

Minor comments:

Other recent work on semantic relatedness during learning may be worth including:

Antony, J. W., & Bennion, K. A. (2022). Semantic associates create retroactive interference on an independent spatial memory task. *Journal of Experimental Psychology: Learning, Memory, and Cognition*, No Pagination Specified-No Pagination Specified. <https://doi.org/10.1037/xlm0001216>

Regarding the repulsion effect of semantic lures – how many semantic lures existed for a particular to-be-learned word pair? Were these included in the stimulus intentionally or were they identified post-

hoc?

"When tested pairs were split into those that were correctly recalled at initial learning (Day 1) and those that were not ..." the authors have language like this in the main text and also in Figure 3 legend, and it's not immediately clear that the statistics and figure only show a contrast of tested/correctly recalled vs restudied. I think rephrasing the figure legend to "When only comparing restudied pairs to pairs that were correctly recalled at initial learning" or something equivalent would solve this. In the main text, it would help to specify that the relatedness x learning interaction is over correctly recalled items before reporting the stats – as written, this info is only explicitly stated when reporting the follow-up t-tests.

"When Fisher z-transformed correlation values were entered into an LMM with word position (cue vs target), learning condition (test vs restudy) and subject identity as random effects, ..." – I believe relatedness is missing here

Figure 4D: Is there a reason why the dots corresponding to subjects are spread across the width of the bars in this figure? It's inconsistent with the other figures and makes it more difficult to inspect the slopes of the subject-specific lines.

Figure 5B: It would be useful to report whether these asymmetric changes significantly differ from 0, as 0 would indicate no changes across learning.

Reviewer #4 (Remarks to the Author):

In this study, the authors address a classic question in memory research, on the roles of semantic relatedness and testing on memory for word pairs. They bring novel approaches to this question that allow them to measure the semantic representation of words before and after learning, and find a number of interesting symmetric and asymmetric changes that occur in the representations of the cue and target words. The paper is well-motivated and connects extensively with the existing literature, and the figures do a nice job of showing their data.

As requested by the editor, I've focused my comments largely on the RSA and imputation procedures. In general I found these approaches to all be statistically appropriate, and the authors' conclusions to be well-supported by the analyses they performed. These novel methods for assessing semantic geometry are an exciting contribution of the paper, in addition to the main scientific claims. I do have a number of specific comments below about some of the details of these analyses and the rationale for how they were selected and designed.

The authors use several different ways of measuring similarity between words. I believe these are: 1) an imputation-based method to get a full similarity matrix (lines 268-278); 2) distance to 20 nearest neighbors, to get a 20-dimension vector for each word that can be used to measure similarity (lines 320-324, 362-366); 3) similarity between participant distance ratings to nearest neighbors and normative semantic distance ratings to these neighbors (lines 371-374). I've organized my comments below to correspond to each of the three metrics:

1) Imputation-based method

After the 4 trials of the "SWAT" task, participants end up making similarity judgments between each word A and 89 of the other 119 words in the experiment. The 30 that are missing include the word B that this word will be (or was) paired with during learning as A-B. The similarity between A and B was intentionally not tested, to ensure that the semantic arrangement task did not influence or become influenced by the episodic associations. In order to estimate this similarity, the authors use the fact that, for 60 of the 89 words for which they *do* have similarity ratings for word A, these words were directly compared to word B. The imputation procedure makes sense conceptually, and the authors

tuned the number of nearest neighbors to provide the best reconstruction of similarities within the observed set of similarities (Supplementary Results, "Validation of Imputation").

a) The exact details of the imputation procedure need to be better described - it appears that the authors are using the `KNNImputer` function from `scikit-learn`, in which case they should describe the options that they used (was the NN weighting set to "uniform" or "distance"?) and should cite the original paper that developed this method (DOI: 10.1093/bioinformatics/17.6.520).

b) The illustration in Fig 2B is confusing to me, because the "To-be-imputed Target" and "Imputed Target" vectors are the same - should the "Imputed Target" vector have no missing values?

c) Was the validation procedure fully comparable to the procedure in the main text? My concern is that the validation procedure removed "a random sample of observed pairwise dissimilarity values" and then attempted to predict them. This makes it sound like random entries in the similarity matrix were removed, whereas the actual imputation process involves having empty "blocks" in the matrix. I.e. If word A and C are in the same 30-word half-list, then the 30 words that word A is missing distances to are the exact same 30 words that C is missing distances to. If the missing distances were totally uncorrelated across words in the validation test, that seems like a substantially easier reconstruction task.

d) An alternative approach here would be to take the 60 words whose distance was measured relative to both word A and word B, and compute the similarity between the 60-dimension vectors of distances to word A and word B. This avoids having to perform any imputation, and seems like it would provide a similar kind of measure. It would be helpful to know why this approach was not used, or whether it yields similar results to the imputation approach (further strengthening confidence in those results).

2) Distance to 20 nearest-neighbors

For the asymmetry analyses (Fig 5), the authors represented each word as a 20-dimension vector based on the distances to its 20 nearest neighbors, and then computed the similarity between these vectors (either initial vs. final representations, or initial word A to final word B and vice versa).

a) I didn't understand the motivation for why this additional approach was needed. The imputation procedure produces a vector representation of each word (e.g. as shown in Fig 2B) during the initial and final phases, which seemingly could have been used for these analyses. Is it important conceptually that only a local neighborhood is considered?

b) Exactly how these 20 values are computed wasn't clear to me. The 20 nearest neighbors were chosen based on word2vec similarity, but then the authors say that "The top 20 largest cosine similarity values, excluding pairs of words where the distance is imputed, were used as the vectorized representation for each word." Are they saying that the word2vec cosine similarity values are the values in this vector? This seems unlikely, because in this case the initial and final representations should be the same. Or is this saying that the top 20 word2vec similarities are used to pick the 20 neighbors, but then the similarity values come from the distances in the SWAT task for those neighbors?

c) Why were 20 nearest neighbors chosen here? It seems like all 60 of the words with measured distances to the associated word-pair could be used (see my suggestion in 1d), which would give a more global representation of semantic meaning.

3) Comparison to normative semantic template

Here, I believe the authors take the 20-dimension vector from the previous approach, and compare it to a 20-dimension vector in which the distances to the 20 neighbors come from the word2vec model.

High similarity between these vectors would therefore indicate that the subject's distances align with the "ground-truth" semantic model.

a) The authors describe this method as measuring "symmetric change within a pair relative to normative semantic space." This phrase "within a pair" seems to imply that this is measuring something about the relationship between the associated words, but my understanding of the method is that it is separately measuring how well word A and word B match their respective normative templates.

b) More confusingly, the abstract states that "Testing additionally repelled semantic representations away from normative semantic space moreso than restudying." However my reading of Figure 6 is that the correlation between the words and the normative space *increased* after testing (Final correlation minus Initial correlation is positive). The authors should clarify how to interpret the sign of the effect here.

4) Minor comments:

a) Line 416 describes Fig 3C/3D by saying "When tested pairs were split into those that were correctly recalled at initial learning (Day 1) and those that were not" but the figure doesn't show separate results for correct vs incorrect pairs. Is this saying that the Figure is just one of these groups (e.g. only the correct pairs)?

b) In Fig 4C the groups are labeled as "To be Learned? No/Yes" but the analysis is showing an Initial vs Final comparison, so the future tense is confusing here. Could this just say learned pairs vs random pairs?

Jun 22, 2023

Dear Reviewers,

We deeply appreciate the enthusiasm about our method and findings, and thank you kindly for your supportive and constructive remarks on our manuscript. Your suggestions have helped us to revise our paper to substantially improve the clarity of our methods, the presentation and narrative flow of our results, and our discussion of the findings and their relationship to the literature. We have also made our writing more concise to adhere to the *Nature Communications* word count guidelines. Direct responses (black text) to each comment from the reviewers (blue text), as well as the line numbers for the corresponding revisions, are provided below.

Sincerely,

Catherine Walsh and Jesse Rissman

Reviewer #1 (Remarks to the Author):

Comments Summary

The authors report some exciting work on how semantic and episodic memory interact with each other, and with different approaches to learning (i.e., re-studying vs. testing) during paired-associate learning. It is encouraging to see work being done on the reciprocal influence of semantic and episodic memory, in addition to the application of interesting statistical approaches in the behavioural experiment space (e.g., representational similarity analysis). In reading the article, some notable questions remain about the overall design of the task which require clarification. There were also some cohesiveness issues in the article as a whole, and in particular sections that are outlined below. In the comments provided, I highlight these outstanding questions and offer some suggestions for improving the writing.

Response: We thank Reviewer #1 for the thoughtful feedback and constructive comments on our work. We deeply appreciate the thoughtful suggestions and are pleased to make the changes that the reviewer has suggested, which we believe have strengthened our paper.

Major Comments

1. You highlight how semantic relatedness can promote memory in the face of interference, but it is also well-documented that semantic knowledge can cause interference, leading to poorer

memory (e.g., proactive interference). Clarify what conditions would lead semantic memory to resolve, rather than cause interference, and explain what is meant by interference in this case (e.g., noisy environment? Competing memories? Etc.). This could also be elaborated on in the discussion.

Response: We thank the reviewer for this insightful point. We have added additional text addressing this relevant literature to our Introduction (lines 54-57).

2. More cohesion was needed in the introduction. It wasn't always clear how each section related to one another, or how they related to the main questions. I think the biggest problem was that the main questions were not introduced early enough. Including these in the first paragraph would be helpful to orient the reader and provide a better structure for the flow of the introduction. For example, the questions that are later outlined in the discussion, were not the main questions that I drew from the introduction. It would also be helpful to introduce the testing effect earlier as a topic of interest, as it was unclear why the authors jumped from talking about semantic relatedness to the testing effect.

Response: We apologize for the overall lack of clarity in our Introduction. As per the editorial guidelines of *Nature Communications*, we kept the discussion of the current work and our main research questions at the end of our Introduction; however, we have restructured the Introduction to discuss the testing effect earlier and make the transitions between topics more clear. We hope that these changes will improve the readability and cohesion of our Introduction.

4. The task needs more clarity in the introduction in order to set up the hypotheses and predictions. It would be helpful to clarify briefly what participants are doing, how the pairs were varied in terms of their semantic relatedness, and what was the difference between the two learning conditions. It was difficult to understand the predictions and keep track of the stimulus manipulations when they weren't well-explained in the introduction.

Response: We thank the reviewer for this suggestion. We absolutely agree that including a more direct description of our task and manipulations lays the groundwork for understanding our predictions. We have now included additional text in the Introduction to this end (lines 102-109).

6. In the methods, the authors show that participants complete two study phases with relatedness judgments, followed by an intermixed restudy/test phase, then the final memory test following a delay. This design raises several questions about the mechanisms behind the findings.

Firstly, the first two study phases are incidental learning tasks with a relatedness judgment. Thinking deeply about the semantic relationship between two words is a much different task

than simply re-typing the word, as in the intermixed block. The differences between re-study vs. test could partially be attributed to shallow vs. deep processing of the pairs. While this is one of the arguments that is typically put forth when explaining the testing effect, the differences could be inflated by this manipulation. Please clarify why it was necessary to change the study method in the intermixed block, rather than having participants complete another relatedness judgment.

Additionally, there may be some effect of switching between a recall task and a typing task in the intermixed block. While this shouldn't have an impact on performance on day 1, given that participants had unlimited time to provide an answer, it could have impacted day 2, particularly for the re-studied items. This effect may occur because the re-tested items may be more difficult to inhibit, particularly when the items are semantically related (again, shallow vs. deep processing). It may be worth including switching as a covariate in the model. In follow up studies, it may also be worth replicating the within-subjects effects, as well as running a between-subjects manipulation where the intermixed block is changed to either a re-study block or a test block depending on the condition.

Last, there is a literature on the benefit of one manipulation over another only when encoding or testing is on mixed list, as it was in this study, than on pure lists (e.g., on the superiority of intentional over incidental learning, in levels of processing paradigms (Craik, JEP :Gen, 2023) and on the memory superiority of emotional over neutral items (Talmi,et al., Journal of Memory and Language, 2007).

Response: We thank the reviewer for these incredibly thoughtful points about our design. Our experimental design decisions to use word re-typing as our restudy condition and to intermix restudy and test trials during the third round of Day 1 learning was based on two primary reasons: First, we wanted to match the behavioral responses across our restudy and test conditions - that is, we wanted to ensure that regardless of the learning condition, participants were always typing out a response. Second, we also sought to create a better match between the intermixed block and the final test, where subjects must complete a cued recall task and type in the retrieved answer. If we had asked subjects to complete an additional relatedness judgment for the restudy condition (rather than re-typing), this might have artificially inflated the testing effect because of episodic differences between encoding and retrieval, as might be explained by theories such as the Transfer Appropriate Processing framework. We have added information to the Methods section (lines 502-507) to clarify this logic.

Regarding the reviewer's second concern about task-switching, we agree that it would hopefully be mitigated by the unlimited time to respond at Day 2. We opted to mix the restudy and test trials based on prior literature on the testing effect that chose an intermixed design (e.g. Halamish and Bjork, 2011; Storm et al., 2014; Rawson et al., 2015, Lehman and Karpicke, 2016). Additionally, intermixing the trials helps to balance the retention interval between the

second learning opportunity (where participants made a relatedness judgment for all pairs) and the third opportunity (where the restudy vs test condition was introduced). Moreover, a meta-analytic review of the testing effect (Rowland, 2014) systematically investigated the effect of list presentation (blocked vs intermixed) and showed that while the average effect size was numerically larger for intermixed design (Hedge's $g = 0.49$) vs blocked design (Hedge's $g = 0.46$ CI), both kinds of designs were able to show testing effects. We also note that the order of word pairs presented in our intermixed block was randomized across participants, so there should not be a systematic effect of task switching that confounds our results.

That said, in an effort to more compellingly rule out a potential impact of task-switching, we have run an additional analysis (now noted in our Methods section, lines 507-510, and reported in detail in our Supplementary Results) where we classified each trial as either a switch trial (i.e. the learning condition differed from the previous trial) or a non-switch trial (i.e. the learning condition was the same as the previous trial), and re-ran two RM-ANOVAs on behavioral accuracy at Day 2 with trial type (switch vs non-switch), relatedness (related vs unrelated) and learning condition (test vs restudy and test (correct at Day 1) vs test (incorrect at Day 1) vs restudy). We found that regardless of whether we split tested trials based on their recall accuracy at Day 1, there was no difference in accuracy at Day 2 based on trial type, and trial type did not interact with our other factors. Based on these results, we believe that task switching did not impact our findings.

We agree that follow up studies with a between-subject manipulation could further elucidate this, in addition to answering interesting questions about the effects of a stable vs. variable learning context.

7. Please explain why the second SWAT task comes after the last memory test and not before. There are pros and cons in both situations, but there could be an additional boost from the testing effect that comes from the retrieval task.

Response: We opted to have the second SWAT protocol come after our final memory test because we were concerned that subjects might practice recall if they encountered words during a SWAT protocol conducted prior to the final test. Because we would have no control over whether such retrieval practice was taking place, nor over which words triggered retrieval and which did not, we feared that this could skew our results on the final memory test and add substantial variability. We agree with the reviewer that it is possible that the retrieval practice on the final memory test might have provided an additional boost to the representational change that we observed in the SWAT data. That said, when we designed our study, we decided that we preferred being able to control the opportunities for recall practice by having each pair systematically retrieved during the final test, rather than doing the SWAT first and not knowing how much participants were recalling paired associates as they encountered each word on the

SWAT. We have added additional text to the Methods section (lines 481-483) to explain our rationale.

8. Clarify whether the word pairs were always presented in the same order (e.g., did “road” always precede “car”?). While forward strength of association was reported, backward strength was not (i.e., how similar is the target to the cue). Would you expect similar representational changes in the second SWAT had the order of the pairs been reversed?

Response: We have clarified in our Methods section that paired words were always presented in the same forward order (lines 515-516). We believe that the effects of reversing the order of the pairs may diverge based on the semantic relatedness of the pair. Popov et al. (2019) showed that testing a word pair (A - B) in the reverse direction (B -> A) only improves later recall in the forward direction (A -> B) if the words in the pair are semantically unrelated. Given that finding, we believe that testing in the reverse direction may have produced the same findings for the semantically unrelated pairs, but likely would have produced no representational change for the semantically related pairs.

9. For recall accuracy it is worth noting how the results were potentially mediated by whether or not the word pairing was correctly recalled. This seems to also be specific to how testing effects and semantics interact with each other. Perhaps you could run something like a mediation analysis or include correctness as a covariate in an LMM.

Response: We absolutely agree that the success of recall is a key factor in our findings - many of our findings are only apparent for the correctly recalled trials. We have clarified in our Methods section (lines 683-684) that final recall success is included as a fixed effect in all of our LMMs and have added more description to each model in the Results section to explicitly describe fixed effects.

10. Figure 5B is slightly confusing, perhaps relabelling the y-axis would be helpful. The way it reads now is that there is asymmetry for both related and unrelated pairs, but just in different ways. In the related pairs, the representation of the cue is moving towards the target, and in the unrelated pairs, the target is shown becoming more similar to the cue representation. You could relabel to something like the following:

0 to -0.3: targets < cues; 0 to 0.3: targets > cues

Response: We thank the reviewer for the suggestion about this figure. We have made the y-axis label more intuitive and have added text to the Results section (lines 228-232) that explains how to interpret the values in the figure.

11. Lines 540-542: To better introduce the next section, briefly describe how these two mechanisms may interact.

Response: We thank the reviewer for this helpful suggestion. We have moved the discussion of these results to the Supplementary Results and have added additional text to describe what we think the underlying mechanism might be.

12. Overall, each section of the results was straightforward, but it wasn't always clear why the next analysis was the logical next step. Similar to comment 11, you could better introduce the next section by highlighting a follow-up questions, for example.

Response: We apologize for the lack of clarity in the transitions in our Results section. We have added additional text throughout the Results section to clarify the logic of the flow of analyses (lines 152-154, 178-181, 241-242).

13. As above, it wasn't clear whether the questions posed in the discussion were as explicit as they needed to be in the introduction. Making the changes to the introduction suggested above will improve the cohesion between the introduction and discussion.

Response: We hope that our implementation of the thoughtful changes suggested by the reviewer has improved the overall cohesion of the Introduction and Discussion sections, and thus enhanced the overall clarity of our manuscript.

14. In addition to the themes that have already been raised in the discussion, another relevant topic is the role of transfer appropriate processing in the testing effect (e.g., Morris, Bransford, & Franks, 1977). In the testing condition, you get the same task in the intermixed phase and test phase which promotes memory by increasing contextual similarity between encoding and retrieval. This is also relevant to the re-study phase, which is more discriminable from the retrieval context.

Response: We thank the reviewer for this suggestion - we have added additional text about the role of transfer appropriate processing to both our Introduction (lines 65-66) and Discussion (lines 319-321).

Minor Comments

1. Line 49: You state that early psychological theories proposed a separation between semantic and episodic memory systems. It is important to state whether this is a psychological distinction, neurological distinction, or both.

Response: We thank the reviewer for this comment - we have rephrased the statement to clarify that the proposed distinctions are both psychological and neurological (line 49).

2. Lines 108-110: Redundant wording at the beginning of the two sentences, you can join them together.

E.g., “This theory proposes that changes in memory strengths are driven by the relative activation of items, such that memory for items that are strongly activated is strengthened, while items that are moderately activated are weakened or differentiated.”

Response: We thank the reviewer for this suggestion - we have made the proposed change (lines 92-94).

3. Evidence from the visual domain is brought up a few times in the introduction, and the way it is phrased makes it seem like a prominent aspect of the research being discussed. To minimize this confusion, you could de-emphasize the visual aspects in your phrasing. For example, you write:

“In the visual domain, recent neuroimaging work has shown asymmetrical integration for novel faces paired with famous faces.”

This could be rephrased as:

“Recent neuroimaging work has shown asymmetrical integration for novel faces paired with famous faces.”

Response: We thank the reviewer for this suggestion - we agree that our previous phrasing made it seem as though we would be more focused on the visual domain than we intended. We have made the changes suggested to the Introduction to de-emphasize the visual nature of the stimuli (lines 83-88).

4. Lines 122-125: Delete “a” from the second last line of the sentence

i.e., “In the present study, we sought to directly investigate the influence of semantic relatedness on the testing effect and understand how episodic paired associate learning might sculpt pre-existing semantic space.”

Response: We thank the reviewer for catching this detail - we have removed the “a”.

5. In setting up the demographics, please also report the average years of education + age range to get a better picture of the sample. The average years of education is of particular interest given its potential link to semantic ability (e.g., more flexible semantic networks).

Response: We have added the age range of our participants to our Methods section (line 432). Unfortunately, we did not collect exact years of education from our subjects. Prolific provides demographic information from pre-screening eligible subjects, which only includes whether or not an individual is currently a student. SONA recruits from the undergraduate population from UCLA; these subjects provided their level of education in a categorical fashion - a high school degree, some college, a bachelor's degree, master's degree or doctorate degree. Of our participants from SONA, all reported either a high school degree (12 years of education) or some college (12-16 years of education) (lines 433-434).

6. In figure 1, it might be helpful to include a legend on the figure to illustrate which items were restudied and which were tested (i.e., red = restudied; blue = tested).

Response: We thank the reviewer for this suggestion - we have adjusted the figure so that the labels for the third learning opportunity are colored to match the conditions in the figure.

7. Lines 449-452: Delete "as" before the word "experienced."

i.e., "...showed greater change in similarity than did a control comparison of arbitrary pairings of words that were never experienced as to-be-learned pairs"

Response: We thank the reviewer for catching this - we have removed the "as".

8. Line 610: Change the wording in this sentence. You say that you will address your first hypothesis, but you have not stated your hypotheses, you have stated your main questions.

Response: We thank the reviewer for this suggestion - we have changed the phrasing to make it more clear that we have stated questions, not explicit hypotheses (line 283).

9. Line 612: "...an approximately 24-hour delay" is a bit awkward, could reword to "after approximately 24 hours."

Response: We thank the reviewer for this suggestion - we have changed the phrasing of this statement.

10. Line 636: Change "effects" to "affects."

Response: We thank the reviewer for catching this - we have changed "effects" to "affects."

11. Lines 664-668: this sentence is very long and does not read clearly, I suggest the following change:

“The Dual Memory Theory proposes that testing creates a directionally specific associative relationship, such that the cue-to-target relationship is strengthened without influencing the backward associative linkage of the target to the cue. This is accomplished by creating an episodic “cue memory” where the cue and target are encoded in the context of a retrieval task, whereas restudying creates a bidirectional association.”

Response: We thank the reviewer for this suggestion - we have changed the phrasing to clarify (lines 317-319).

Reviewer #2 (Remarks to the Author):

This is an important contribution to the literature on the interplay between episodic and semantic memory, demonstrating how semantic representational space is potentially reshaped by episodic learning. Using a novel behavioural representational similarity analysis approach, the authors reveal differential reorganisation of semantic space based on the degree to which cue and target representations are related, with differences observed depending on the learning condition. These findings suggest that episodic learning might distort semantic space in an adaptive manner by strengthening associations within pairs and potentially reducing interference from lures.

Overall, I found this to be a very well-executed study. The authors are to be commended for developing an elegant behavioural paradigm and their meticulous attention to detail. The manuscript itself is extremely well-written; hypotheses are well articulated, and findings are interpreted in an appropriate and measured way. I have only very minor comments that the authors might like to address, on what is otherwise a fine piece of work.

Response: We thank Reviewer #2 for the enthusiastic appraisal of our work. We are grateful for the helpful comments and believe that the resulting changes have improved our paper.

Comments:

I wondered if there is a possibility that some of these words are more salient/memorable to participants over others?

Response: We thank the reviewer for this interesting comment. We have conducted a new analysis to formally examine this. We operationalized memorability as the average recall accuracy (across participants) at Day 2 for the target word of each pair. We show that our word

pairs have a wide range of memorability (ranging from pairs that only 5% of subjects recalled to pairs that 91% of subjects recalled). But more importantly, we also show that memorability does not significantly differ across our word sets (which were used to randomize the testing vs restudying manipulation), so the memorability of the word pairs should not confound our results. We have included these results in our Methods section (lines 519-523), Expanded Data Figure 4 and Supplementary Results.

Does education play a role here or other individual differences?

Response: As stated in our response to Reviewer #1, we unfortunately did not collect this information. Prolific provides demographic information from pre-screening eligible subjects, which only includes whether or not an individual is currently a student. SONA recruits from the undergraduate population from UCLA; these subjects provided their level of education in a categorical fashion - a high school degree, some college, a bachelor's degree, master's degree or doctorate degree. Of our participants from SONA, all reported either a high school degree (12 years of education) or some college (12-16 years of education). We have reported this information in the Methods section (lines 433-434). We regret that our rather limited demographic info precludes us from looking for potential individual differences related to education.

Did the authors collect any information regarding the time taken to organise the words during the SWAT task?

Response: As reported in the Methods section of our main text (line 543), SWAT trials took a median duration of 7.14 minutes each.

On page 8 the authors mention that word pairs that were incorrectly recalled at Day 1 were excluded from the analyses. How many of these pairs were excluded?

Response: The number of pairs excluded because they were incorrectly recalled at Day 1 varied across subjects, with a mean number of pairs = 11.23 (SD = 4.57 pairs). We have added this information to the Methods section (lines 595-596).

I would have liked to have seen further details regarding the exploratory analyses where semantic lures were identified based on their associated to-be-learned targets. How many pairs were tested and how were these proximities identified? It was not entirely clear to me how this was conducted.

Response: We apologize to the reviewer for the lack of clarity about this. As we reviewed the details of this analysis scheme to clarify the Methods section, we became concerned it may have

been vulnerable to potential regression-to-the-mean effects. We had defined the lures as pairs that were particularly close to a given cue word based on the Day 1 SWAT measurement, so our finding that lures were repelled away from to-be-learned cue words after learning may have been confounded by how they were defined. We ultimately decided it would be more rigorous to define the strength of the lures based on LSA cosine similarity - an unbiased measure of semantic distance, which we initially used to select our semantically related and unrelated to-be-learned pairs. We are enthusiastic about this revised approach, as the results (Fig 5C) are exactly in line with what we believe would be predicted by the Non-Monotonic Plasticity Hypothesis (NMPH). Specifically, we found almost no representational change for weak/non-lures, whereas very strong lures were drawn closer to cues regardless of learning condition (as there likely was co-activation of the strongly semantically related concepts during learning). Most interestingly, we found that the representations of moderate strength lures were repelled away from their associated cue words in semantic space, but only for pairs that underwent testing. This effect is not only consistent with the predictions of NMPH, but also with an emerging body of work showing that competition adaptively distorts and repels overlapping episodic representations so they become less similar (Hulbert and Norman, 2015; Chanales et al., 2021; Drascher and Kuhl, 2022).

We have added additional text to the Methods section describing the new analysis (lines 709-738), but in brief, we first identified all possible 118 potential lures for each cue word (all other words in our corpus, except for the to-be-learned target pair - 7,080 potential pairs for each subject). We additionally excluded lure pairs that were associated with a to-be-learned pair that was incorrectly recalled at Day 2, as the incorrect responses were often other words in our set (i.e. other potential lure pairs), and the retrieval of this (incorrect) pair in the final text prior to performing the SWAT protocol may act as a confound to our analysis.

Once we identified the lure pairs for each subject, we classified each into a lure strength bin (weak/non-lure, moderate lure, strong lure, very strong lure) depending on the LSA cosine similarity of the to-be-learned cue word to the potential lure. We provide concrete examples of each kind of lure (lines 718-720) and Expanded Data Figure 5 to clarify how semantically close each potential lure is to the to-be-learned cue.

Finally, we used a linear mixed-effects model to test whether the change in similarity of a given to-be-learned cue/lure pair depended on the strength of the lure and the relatedness and learning condition of the associated to-be-learned pair.

The exploratory analyses, while very interesting, quickly becomes difficult to follow. I wonder if it is possible to include representative examples to help the reader follow exactly what is being tested in each section. By providing representative examples of semantically related and

unrelated pairs, it may help to contextualise the research for readers who are not specialists in this field.

Response: We thank the reviewer for this helpful suggestion! To improve clarity, we have included a concrete representative example of the word pair “GENDER - FEMALE” to illustrate each of our analyses of representational change (lines 210-213, 223-226).

Minor typographical errors:

Figure 3 caption: “as a function semantic relatedness”
[“of” is missing]

Line 452: “never as experienced”
[remove “as”]

Line 458: “sematic”
[should read “semantic”]

Response: We thank the reviewer for catching these typos - we have made the appropriate changes in the main text.

Reviewer #3 (Remarks to the Author):

Summary

In this paper, Walsh and Rissman develop a novel spatial arrangement task (SWAT) to probe how learning word pairs of varying relatedness impacts the organization of participants’ semantic knowledge, and how this learning is modulated by testing versus restudying the pairs. This task was developed to infer relatedness scores for pairs of words that were studied together. They find that related pairs of words benefit less from the testing effect relative to unrelated pairs (less of a difference in accuracy for tested versus restudied pairs), and using the SWAT, they find that word pairs are represented more similarly after learning, are repulsed from unlearned semantic neighbors (i.e. semantic lures), and that there are asymmetries in whether the target or cue words are impacted by learning. Finally, they find that semantic relatedness exerts opposing forces on how the learning-related changes in a target words’ semantic representation relates to its later. The authors interpret their findings through the lens of the non-monotonic plasticity hypothesis and other computational frameworks.

Evaluation

Spatial arrangement tasks have long been used to infer semantic relatedness, but the imputation approach is an incredibly clever modification of the task that enables the authors assess the relatedness of word pairs without drawing attention to them or evoking retrieval strategies that may not necessarily reflect their semantic organization. I can imagine many other memory-related questions that would benefit from this task, and the many analyses that employ this task result in a rich dataset that can target different ways in which associations in memory can change. The question of the testing effect and how it is impacted by semantic similarity will be of interest to memory researchers as there has been a renewed focus on interactions between episodic and semantic memory in recent years. As you can see below, my main suggestions involve improving the clarity of the paper and expanding on the theoretical motivation and implications of the results.

Response: We thank Reviewer #3 for the positive evaluation of our work. The thoughtful comments from the reviewer were incredibly helpful, and we believe that the changes have made our paper more clear and better situated in the existing literature.

Major comments

The major novelty of the paper is the development of the SWAT and the imputation approach that is used to characterize changes in how specific words were associated with each other and how their relationship to other words changed after learning. However, there is a long history of studying how recent/episodic experiences shape or update semantic knowledge that is not referenced here. Lots of examples can be found in the below papers for example.

Yee, E., & Thompson-Schill, S. L. (2016). Putting concepts into context. *Psychonomic Bulletin & Review*, 23(4), 1015–1027. <https://doi.org/10.3758/s13423-015-0948-7>

Casasanto, D., & Lupyan, G. (2015). All Concepts are Ad Hoc Concepts. In E. Margolis & S. Laurence (Eds.), *The Conceptual Mind: New directions in the study of concepts* (pp. 543–566). MIT Press.

Connell, L., & Lynott, D. (2014). Principles of Representation: Why You Can't Represent the Same Concept Twice. *Topics in Cognitive Science*, 6(3), 390–406. <https://doi.org/10.1111/tops.12097>

Estes, Z. Attributive and relational processes in nominal combination. *Journal of Memory and Language* 48, 304-319, doi:10.1016/S0749-596X(02)00507-7 (2003).

Solomon, S. H. & Thompson-Schill, S. L. Finding features, figuratively. *Brain and Language* 174, 61-71 (2017)

Response: We thank the reviewer for their suggestions here, particularly for including helpful citations! We agree that this literature is important for framing the bidirectional interactions between episodic and semantic memory and have included additional text, including most of these citations, in the Introduction (lines 54-59) on these topics.

There are two topics that would strengthen the paper. The first is a discussion of the SWAT/imputation approach and its advantages/disadvantages relative to other measures of semantic relatedness such as semantic priming, free association, spatial arrangement without imputation, etc.

Response: We thank the reviewer for this suggestion - we absolutely agree that it is important to frame our novel methodology in reference to existing methods so it is more clear what added benefits our specific approach offers. To that end, we have added text to the Discussion (lines 383-390) about existing approaches for indexing semantic relatedness and how our SWAT/imputation approach diverges from them.

The second is more of a theoretical discussion on the dynamic nature of semantic memory and how new learning can change and update semantic knowledge networks. For example, does exposure to GENDER – FEMALE decrease the path length between these two words, create a direct path that previously was only indirect, or expand the meaning of gender to include more ‘female’ features? Would testing or restudying affect the way that the semantic network is being changes? These are purely speculative questions but are fascinating to think about!

Response: We agree that they are fascinating questions! We suspect it may be a little bit of all three potential mechanisms, depending on the characteristics of the word. For example, we suspect that for semantically related pairs, learning might decrease the path length between the words, resulting in the words being pulled closer together in semantic space (as we saw in our pairwise similarity analyses), and that learning may sculpt the meaning of the cue word to highlight shared (previously existing) features, although our analyses of how words change relative to normative space suggest that this may not occur through the addition of completely novel features. It is possible that this sculpting may create a path connecting the cue and target words where there wasn't one previously, as we see in our analysis of the pairwise representational change of lures, where the close semantic associates to the cue word are drawn closer after learning. Our results in this analysis also suggest that there may also be a process where testing (but not restudying) pushes moderately related lure pairs (i.e. words that may interfere with learning of the correct to-be-learned pairing) away in semantic space, suggesting

that there is a differential impact of learning condition on the changes to overall semantic space. We have added additional text to the Discussion (lines 342-356) to highlight these questions.

The analyses reported are complex and there are times where a clearer description of the methods could greatly increase the readability of the Results section. Since each analysis is subtly different, it can be hard to keep track of the similarities and differences across the sections. It might be useful to start each section with an example – I found the example describing the semantic lures and to-be-learned pairs very helpful, and continuing that example throughout the paper might help to clarify and differentiate the many complex analyses that are reported here. For instance, the section on changes in semantic space could be well served by listing 5 of the 20 nearest neighbors that the vector for GENDER was compared to, etc. It might also be useful to have either a separate analysis approach figure showing what is being correlated when (cue to target, cue to target before learning, target to cue before learning, target to 20 neighbors, etc), or an equivalent schematic that visualizes and summarizes the main findings of the paper.

Response: We apologize for the lack of clarity. To improve the readability of the Results section, we have added additional transitions to highlight the logic of the flow of analyses (lines 152-154, 178-181, 241-242) and taken our concrete example of GENDER-FEMALE and extended it to the other analyses to make it more clear what each analysis is investigating (lines 210-213, 223-226). We have also created an additional figure (Figure 4) that visually depicts what is being compared for each analysis.

One drawback of excluding forgotten items from the retest condition on Day 1 is that the comparison to restudying is confounded, because items that are restudied on Day 1 will include a mixture of items that would have been remembered or forgotten if tested (both if tested in place of restudy, or if tested after a chance to restudy). This means that memory will pretty much guaranteed to be worse in the restudy condition relative to the test condition. I believe it's more common to compare all tested trials with all restudied trials and this is the approach that was pre-registered. This is a fairly large deviation from the pre-registration and it's worth mentioning it briefly in the Results section, not just in the discussion and supplement.

Response: We agree with the reviewer that this is a potential confound - however, we opted for this procedure because we wanted to be able to make claims about the effects of testing over restudying. We show in Figure 3B the results for all pairs, which demonstrates a clear benefit for tested pairs, regardless of the recall success at Day 1. A revised version of Figure 3C goes on to show that the behavioral advantage of testing seen in Figure 3B is heavily driven by tested pairs that were successfully recalled at Day 1. We opted not to provide corrective feedback to participants at Day 1, meaning that pairs that were not successfully retrieved at Day 1 are also much more likely to show failed recall on Day 2.

Moreover, previous literature (Kornell et al., 2011; Halamish and Bjork, 2011; Storm et al., 2014) has suggested the benefits of testing only occur when a tested item is successfully retrieved. In contrast, even though not every restudied item would be recalled if they had been tested, they all underwent successful restudying (in that there was an opportunity to restudy them all), so we still could investigate the difference between “successful” testing and restudying.

Although we did pre-register that we would investigate the behavioral accuracy at Day 2 when splitting the tested pairs into those correctly recalled at Day 1 and those incorrectly recalled, we acknowledge that our pre-registered analyses of representational change did not state an intention to exclude pairs that were incorrectly recalled at Day 1. We ultimately opted to exclude them for both theoretical reasons (as described above, these pairs should not get the benefit of testing, since they were not successfully retrieved) and for statistical power reasons (we ultimately did not have enough pairs in that condition to make any statistically valid claims about the representational status of such pairs). Despite our rationale, we acknowledge that this choice was a deviation from our pre-registration. To that end, we have followed the reviewer’s suggestion to add additional language in our Results section (lines 167-168) and Methods section (lines 594-597) to make that clear.

Figure 3: The A and B panels might be clearer if they were grouped by day instead of relatedness, with relatedness on the x-axis instead of day. Then the Day 2 panel would reflect all components of the 2x2 ANOVA that is the first analysis presented in the Methods. It would also be a visual reminder that the Restudy data from Day 1 is never meaningfully compared to Day 2 since it is from a different task.

Response: We thank the reviewer for the helpful suggestion. As suggested, we have re-grouped Figure 3A and B by day, rather than by relatedness.

Word2vec model predicting the change in semantic relatedness: I personally don’t see the value of including this in the main text unless there is a deeper discussion of how and why these results diverge from those observed with the SWAT. They may fit better in the SI.

Response: We thank the reviewer for this suggestion - we have moved this analysis to the Supplementary Information.

Regardless of where it ends up though, according to the Methods, this model included “the semantic relatedness of the pair, learning condition, recall success at final test and word’s position in the to-be-learned pair” – but only the effect of learning condition is reported in the

Results. All tested predictors should be reported here, even if combined, e.g. No other predictors were significant (all t 's < X, a p 's > 0.XX).

Response: We thank the reviewer for catching this oversight - we have included a statement about the rest of the predictors in the Results section (lines 172, 196, 269).

Minor comments:

Other recent work on semantic relatedness during learning may be worth including:

Antony, J. W., & Bennion, K. A. (2022). Semantic associates create retroactive interference on an independent spatial memory task. *Journal of Experimental Psychology: Learning, Memory, and Cognition*, No Pagination Specified-No Pagination Specified.

<https://doi.org/10.1037/xlm0001216>

Response: We thank the reviewer for this relevant citation. We have included it in our Introduction in our discussion of the bidirectional influence of episodic and semantic memory (lines 51-58).

Regarding the repulsion effect of semantic lures – how many semantic lures existed for a particular to-be-learned word pair? Were these included in the stimulus intentionally or were they identified post-hoc?

Response: We thank the reviewer for this suggestion - as described in a response to Reviewer #2, we have included additional text in our Methods section better describing this analysis (lines 709-738). In summary, we identified all possible 118 potential lures for each cue word (all other words in our corpus, except for the to-be-learned target pair - 7,080 potential pairs for each subject) and then excluded lure pairs were associated with a to-be-learned pair that was incorrectly recalled at Day 2, as the incorrect responses were often other words in our set (i.e. other potential lure pairs), and the retrieval of this (incorrect) pair in the final test prior to performing the SWAT protocol may act as a confound to our analysis. As such, there was a varying number of semantic lures for each pair of words for each participant, as we excluded lures associated with to-be-learned pairs that were not correctly recalled at the Day 2 final test. This resulted in 1652 - 5900 lures per subject.

“When tested pairs were split into those that were correctly recalled at initial learning (Day 1) and those that were not ...” the authors have language like this in the main text and also in Figure 3 legend, and it’s not immediately clear that the statistics and figure only show a contrast of tested/correctly recalled vs restudied. I think rephrasing the figure legend to “When only comparing restudied pairs to pairs that were correctly recalled at initial learning” or something

equivalent would solve this. In the main text, it would help to specify that the relatedness x learning interaction is over correctly recalled items before reporting the stats – as written, this info is only explicitly stated when reporting the follow-up t-tests.

Response: We thank the reviewer for this helpful suggestion. We have updated the text in the figure caption, in addition to displaying Day 2 recall performance for pairs that were tested/incorrectly recalled at Day 1, to highlight that our primary interaction effect is with the tested/correctly recalled condition.

“When Fisher z-transformed correlation values were entered into an LMM with word position (cue vs target), learning condition (test vs restudy) and subject identity as random effects, ...” – I believe relatedness is missing here

Response: We thank the reviewer for their attention to detail - this specific statement was referring to the random slopes that were included in the LMM, which did not include relatedness. We now add additional clarification about each of our LMMs in our Results section (lines 158-160, 185-187, 213-215, 232-233) to make it more clear what fixed effects and random effects were included in each model.

Figure 4D: Is there a reason why the dots corresponding to subjects are spread across the width of the bars in this figure? It’s inconsistent with the other figures and makes it more difficult to inspect the slopes of the subject-specific lines.

Response: We have adjusted this figure to be consistent with our updated analysis and ensured that it is stylistically consistent with the rest of the figures in our manuscript.

Figure 5B: It would be useful to report whether these asymmetric changes significantly differ from 0, as 0 would indicate no changes across learning.

Response: We thank the reviewer for this suggestion - we have run the additional one-sample t-tests comparing each of the asymmetry values to zero, where we show that for unrelated pairs, there was no difference relative to 0 ($t_{(78)}=0.814$, $p=0.418$, $d = 0.18$). In contrast, while related pairs showed a numerically negative asymmetry value with a moderate effect size, this result was only a non-significant trend after corrections for multiple comparisons ($t_{(79)}=2.157$, $p=0.068$, $d=0.49$). We have included the outcomes in the Results section (lines 235-238) and updated the figure accordingly.

Reviewer #4 (Remarks to the Author):

In this study, the authors address a classic question in memory research, on the roles of semantic relatedness and testing on memory for word pairs. They bring novel approaches to this question that allow them to measure the semantic representation of words before and after learning, and find a number of interesting symmetric and asymmetric changes that occur in the representations of the cue and target words. The paper is well-motivated and connects extensively with the existing literature, and the figures do a nice job of showing their data.

As requested by the editor, I've focused my comments largely on the RSA and imputation procedures. In general I found these approaches to all be statistically appropriate, and the authors' conclusions to be well-supported by the analyses they performed. These novel methods for assessing semantic geometry are an exciting contribution of the paper, in addition to the main scientific claims. I do have a number of specific comments below about some of the details of these analyses and the rationale for how they were selected and designed.

Response: We thank Reviewer #4 for the careful evaluation of our methods and this enthusiastic appraisal of the overall rigor and soundness of our approach. We are grateful that the reviewer agrees that our novel methodology for indexing semantic relatedness is a particularly exciting contribution of this paper.. The reviewer's thoughtful comments have helped us to clarify details of our procedure and make our manuscript stronger overall.

The authors use several different ways of measuring similarity between words. I believe these are: 1) an imputation-based method to get a full similarity matrix (lines 268-278); 2) distance to 20 nearest neighbors, to get a 20-dimension vector for each word that can be used to measure similarity (lines 320-324, 362-366); 3) similarity between participant distance ratings to nearest neighbors and normative semantic distance ratings to these neighbors (lines 371-374). I've organized my comments below to correspond to each of the three metrics:

1) Imputation-based method

After the 4 trials of the "SWAT" task, participants end up making similarity judgments between each word A and 89 of the other 119 words in the experiment. The 30 that are missing include the word B that this word will be (or was) paired with during learning as A-B. The similarity between A and B was intentionally not tested, to ensure that the semantic arrangement task did not influence or become influenced by the episodic associations. In order to estimate this similarity, the authors use the fact that, for 60 of the 89 words for which they *do* have similarity ratings for word A, these words were directly compared to word B. The imputation procedure makes sense conceptually, and the authors tuned the number of nearest neighbors to provide the best reconstruction of similarities within the observed set of similarities (Supplementary Results, "Validation of Imputation").

a) The exact details of the imputation procedure need to be better described - it appears that the authors are using the KNNImputer function from scikit-learn, in which case they should describe the options that they used (was the NN weighting set to "uniform" or "distance"?) and should cite the original paper that developed this method (DOI: 10.1093/bioinformatics/17.6.520).

Response: We apologize for this omission - we have added the appropriate citation and information about the procedure to our Methods section (lines 566-568).

b) The illustration in Fig 2B is confusing to me, because the "To-be-imputed Target" and "Imputed Target" vectors are the same - should the "Imputed Target" vector have no missing values?

Response: We thank the reviewer for this clarifying suggestion. We had intended to show, for illustrative purposes, what the imputation process would be like for a single pairwise similarity value (emphasized with the black outline of a single value across all vectors in the figure), rather than all missing values at once (which is what occurs in the true imputation process). We have adjusted the figure to better reflect our true imputation process, where – as the reviewer correctly notes – the final imputed vector has no missing values.

c) Was the validation procedure fully comparable to the procedure in the main text? My concern is that the validation procedure removed "a random sample of observed pairwise dissimilarity values" and then attempted to predict them. This makes it sound like random entries in the similarity matrix were removed, whereas the actual imputation process involves having empty "blocks" in the matrix. I.e. If word A and C are in the same 30-word half-list, then the 30 words that word A is missing distances to are the exact same 30 words that C is missing distances to. If the missing distances were totally uncorrelated across words in the validation test, that seems like a substantially easier reconstruction task.

Response: We thank the reviewer for their attention to detail and for this insightful question. We did in fact remove random entries in the similarity matrix; however, we removed them from the initial, unimputed matrix. Doing this ensured that the difficulty of the reconstruction was matched in our validation and true imputation procedure. We have clarified this and added additional information in our Supplementary Methods and Results sections.

d) An alternative approach here would be to take the 60 words whose distance was measured relative to both word A and word B, and compute the similarity between the 60-dimension vectors of distances to word A and word B. This avoids having to perform any imputation, and seems like it would provide a similar kind of measure. It would be helpful to know why this

approach was not used, or whether it yields similar results to the imputation approach (further strengthening confidence in those results).

Response: We thank the reviewer for this interesting idea! We initially did explore this approach when developing our procedure, but in preliminary pilot data and methods development, we found that the reconstruction from this approach was slightly less accurate than an imputation approach. That said, we have computed similarity values for each of our imputed word pairs and found a positive correlation between this shared vector approach and our imputation approach (initial arrangement: $r = 0.63$, final arrangement: $r = 0.71$). These results, further validating our imputation approach, are reported in the Supplementary Information. Further studies might explicitly test these approaches against one another and determine whether there are boundary conditions where one method might be more accurate than the other.

2) Distance to 20 nearest-neighbors

For the asymmetry analyses (Fig 5), the authors represented each word as a 20-dimension vector based on the distances to its 20 nearest neighbors, and then computed the similarity between these vectors (either initial vs. final representations, or initial word A to final word B and vice versa).

a) I didn't understand the motivation for why this additional approach was needed. The imputation procedure produces a vector representation of each word (e.g. as shown in Fig 2B) during the initial and final phases, which seemingly could have been used for these analyses. Is it important conceptually that only a local neighborhood is considered?

Response: We agree with the reviewer that we could have used the entire 60-dimension vector for these analyses; however, we opted to use the local neighborhood because we wanted to exclude words that were very semantically distant from the target word, as these distant words would share few features or associates. According to the Non-Monotonic Plasticity Hypothesis (Ritvo et al., 2019), representational change is driven by the relative co-activation of shared features, and when there is a relatively low co-activation, there is no representational change. As such, we did not predict that semantically distant pairs of words would show any representational change, and believed that including these pairs of words would obscure our hypothesized effects in the local semantic neighborhood. We have added additional text explaining this logic to the Methods section (lines 623-628).

b) Exactly how these 20 values are computed wasn't clear to me. The 20 nearest neighbors were chosen based on word2vec similarity, but then the authors say that "The top 20 largest cosine similarity values, excluding pairs of words where the distance is imputed, were used as the vectorized representation for each word." Are they saying that the word2vec cosine similarity values are the values in this vector? This seems unlikely, because in this case the initial and final

representations should be the same. Or is this saying that that the top 20 word2vec similarities are used to pick the 20 neighbors, but then the similarity values come from the distances in the SWAT task for those neighbors?

Response: We apologize for the confusion about our methods. As the reviewer stated above, we did indeed use the top 20 word2vec similarities to pick the neighbors, and the vectorized representation of each word came from the SWAT distances for those neighbors. We have added text to the Methods section (lines 628-631) to clarify this procedure.

c) Why were 20 nearest neighbors chosen here? It seems like all 60 of the words with measured distances to the associated word-pair could be used (see my suggestion in 1d), which would give a more global representation of semantic meaning.

Response: As we stated in our response to point 2a, we restricted our analysis to a local semantic neighborhood because we believed that this was where we would see our hypothesized representational change, and that including the all measured words would include pairs with few shared features and thus not show representational change, which might obscure our hypothesized effects.

3) Comparison to normative semantic template

Here, I believe the authors take the 20-dimension vector from the previous approach, and compare it to a 20-dimension vector in which the distances to the 20 neighbors come from the word2vec model. High similarity between these vectors would therefore indicate that the subject's distances align with the "ground-truth" semantic model.

a) The authors describe this method as measuring "symmetric change within a pair relative to normative semantic space." This phrase "within a pair" seems to imply that this is measuring something about the relationship between the associated words, but my understanding of the method is that it is separately measuring how well word A and word B match their respective normative templates.

Response: We thank the reviewer for this suggestion. We agree this wording was clumsy and misleading to readers. As per the suggestion of Reviewer #3, we have moved this analysis to the Supplementary Methods and Results. We have also changed our description of the method to be more clear that the change is symmetric in that both words change relative to the normative semantic space, rather than relative to each other, and added text to the Discussion section to make the description of the results more clear (lines 350-356).

b) More confusingly, the abstract states that "Testing additionally repelled semantic representations away from normative semantic space moreso than restudying." However my

reading of Figure 6 is that the correlation between the words and the normative space *increased* after testing (Final correlation minus Initial correlation is positive). The authors should clarify how to interpret the sign of the effect here.

Response: We sincerely apologize for this confusion. We have moved this analysis to the Supplementary Results, removed the line from the abstract and added text in the Discussion section (lines 350-356) to more clearly describe the findings.

4) Minor comments:

a) Line 416 describes Fig 3C/3D by saying "When tested pairs were split into those that were correctly recalled at initial learning (Day 1) and those that were not" but the figure doesn't show separate results for correct vs incorrect pairs. Is this saying that the Figure is just one of these groups (e.g. only the correct pairs)?

Response: We were indeed only showing the results for the tested pairs that were correctly recalled at Day 1 in Figure 3. As we discussed in our response to Reviewer #3, we have updated the text in the figure caption to make this comparison more clear, in addition to displaying the tested/incorrectly recalled at Day 1 to highlight that our primary interaction effect is with the tested/correctly recalled condition.

b) In Fig 4C the groups are labeled as "To be Learned? No/Yes" but the analysis is showing an Initial vs Final comparison, so the future tense is confusing here. Could this just say learned pairs vs random pairs?

Response: We thank the reviewer for this suggestion - we have changed the label of the figure to make it more clear.

REVIEWERS' COMMENTS

Reviewer #1 (Remarks to the Author):

I am very satisfied with the revision and have no further comments. I commend the authors on the excellent job they did in addressing the reviewers' comments.

Reviewer #2 (Remarks to the Author):

The authors have responded in a thoughtful and careful manner to all of my comments. I am happy with the revised manuscript and believe this paper will make a very nice contribution to the literature.

Reviewer #3 (Remarks to the Author):

The authors have addressed my points and the manuscript is greatly improved. I look forward to seeing this manuscript in published form!

Reviewer #4 (Remarks to the Author):

The authors have addressed all of my major concerns in their revision, and these changes have further improved my positive assessment of the approach and results of the paper.

One minor comment, now that more details have been provided about the imputation procedure: was the distance matrix symmetrized after imputation? The KNNImputer function is treating the rows of the distance matrix as just generic feature vectors, so it seems that the imputed distance matrix will not be exactly symmetric (since the imputation procedure does not know that $\text{dist}[a,b]$ and $\text{dist}[b,a]$ correspond to the same dissimilarity). If the matrix is not being symmetrized, which of the two distance values is used to define the dissimilarity between two items?

Reviewer #1 (Remarks to the Author):

I am very satisfied with the revision and have no further comments. I commend the authors on the excellent job they did in addressing the reviewers' comments.

Response: We thank Reviewer #1 for their kind words and helpful feedback throughout the review process.

Reviewer #2 (Remarks to the Author):

The authors have responded in a thoughtful and careful manner to all of my comments. I am happy with the revised manuscript and believe this paper will make a very nice contribution to the literature.

Response: We thank Reviewer #2 for their helpful feedback and kind words about our manuscript.

Reviewer #3 (Remarks to the Author):

The authors have addressed my points and the manuscript is greatly improved. I look forward to seeing this manuscript in published form!

Response: We thank Reviewer #3 for their helpful commentary and their excitement about our manuscript.

Reviewer #4 (Remarks to the Author):

The authors have addressed all of my major concerns in their revision, and these changes have further improved my positive assessment of the approach and results of the paper.

One minor comment, now that more details have been provided about the imputation procedure: was the distance matrix symmetrized after imputation? The KNNImputer function is treating the rows of the distance matrix as just generic feature vectors, so it seems that the imputed distance matrix will not be exactly symmetric (since the imputation procedure does not know that $\text{dist}[a,b]$ and $\text{dist}[b,a]$ correspond to the same dissimilarity). If the matrix is not being symmetrized, which of the two distance values is used to define the dissimilarity between two items?

We thank the reviewer for this suggestion. We have included additional text (lines 600-601) which specify that we did not symmetrize the distance matrix and only used the upper triangle for our analyses.